# Bioprocessing and Screening of Indigenous Wastes for Hyper Production of Fungal Lipase

Usman Ali [1], Zahid Anwar [1,*], Shoaib Hasan [1], Muddassar Zafar [1], Noor ul Ain [1], Fareed Afzal [2], Waseem Khalid [3,*], Muhammad Abdul Rahim [2], Hanae Naceiri Mrabti [4], Ammar AL-Farga [5] and Hamdi Abdel Rasool Abdelsamad Eljeam [6]

[1] Department of Biochemistry and Biotechnology, University of Gujrat, Gujrat 50700, Pakistan; alisher3484334@gmail.com (U.A.); shoaibhasan9293600@gmail.com (S.H.); noorulainbutt885@gmail.com (N.u.A.); muddassar.zafar@uog.edu.pk (M.Z.)

[2] Department of Food Science, Faculty of Life Sciences, Government College University, Faisalabad 38000, Pakistan; fareedansarifsn@gmail.com (F.A.); abdul.rahim@gcuf.edu.pk (M.A.R.)

[3] University Institute of Food Science and Technology, The University of Lahore, Lahore 54000, Pakistan

[4] High Institute of Nursing Professions and Health Techniques of Casablanca, Casablanca 20260, Morocco

[5] Department of Biochemistry, College of Science, University of Jeddah, Jeddah 21577, Saudi Arabia; amalfarga@uj.edu.sa

[6] Department of English, College of Science and Arts, King Khalid University, Abha 61421, Saudi Arabia

* Correspondence: zahid.anwar@uog.edu.pk (Z.A.); waseem.khalid@uifst.uol.edu.pk (W.K.)

**Abstract:** Background: Lipase is one of the most important enzymes produced from microbial fermentation. Agricultural wastes are a good source of enzyme production because they are cost-effective and production rates are also higher. Method: In this study, eight lignolitic substrates were screened for lipase production. Results: Out of these substrates, guava leaves showed maximum activity of 9.1 U/mL from *Aspergillus niger* by using the solid-state fermentation method. Various factors such as temperature, pH, incubation period, moisture content, inoculum size, and substrate size that influence the growth of fungi were optimized by response surface methodology (RSM), and then characterization was performed. When all physical and nutritional parameters were optimized by RSM, the maximum lipase activity obtained was 12.52 U/mL after 4 days of incubation, at pH 8, 40 °C temperature, 3 mL inoculum size, 20% moisture content, and 6 g substrate concentration. The enzyme was partially purified through 70% ammonium sulfate precipitation. After purification, it showed 34.291 U/mg enzyme activity, increasing the purification fold to 1.3. The enzyme was then further purified by dialysis, and the purification fold increased to 1.83 having enzyme activity of 48.03 U/mg. Furthermore, activity was increased to 132.72 U/mg after column chromatography. A purification fold of 5.07 was obtained after all purification steps.

**Keywords:** lipase; solid-state fermentation; RSM; purification

## 1. Introduction

Enzymes are used worldwide for clean and dynamic industrial processes and have an important contribution to the development of sustainable industrial procedures [1]. Enzymatic processes are more beneficial than traditional chemical processes, including reaction conditions [2], higher specificity, and higher selectivity, leading to better resource utilization and lower production of waste products [3]. Enzymes have earned worldwide recognition for their extensive use in many industries, e.g., medicine, chemistry, energy production, and agriculture. Processes involving the use of enzymes are swiftly adopted due to the decrease in processing time, less utilization of energy, low cost, and nontoxic and environment-friendly properties [4]. Other major factors that promote the use of microbial enzymes in industrial applications are increasing consumer goods demand, depletion of natural assets, and protection of the environment as they are eco-friendly [5]. Enzymes

require normal temperature and pressure to react and are used as a substitute for dangerous chemicals due to their nontoxic and noncorrosive nature [6].

Lipases have emerged as leading biocatalysts in recent years and account for almost ten percent of the enzyme market [7]. Because of the massive utilization of fossil fuels in vehicular applications, they are running out of stock. It is therefore imperative that environment-friendly and renewable alternatives of energy, such as biodiesel, are generated in a less expensive way. Lipases are very important because they can hydrolyze acyl glycerol into glycerol and fatty acids and affect other reactions, such as esterification and transesterification [8]. Lipases are further categorized into monoglycerides, diglycerides, and fatty acids such as serine hydrolases that breakdown triglycerides, have very less water solubility, and where reactions take place at the interface of lipid–water [9]. Lipase has a hydrophobic lid that is essential for its interfacial function [10],and this process is known as interfacial activation. Lipases catalyze esterification, transesterification, and interesterification reactions in organic medium [11].

Lipase differs greatly in its character and is widespread in animals, plants, and microorganisms [12]. Microbial lipases have received a great deal of industrial attention than those obtained from plants and animals as they can operate under severe conditions such as high temperature, have stability in organic solvents, chemoselectivity, and enantioselectivity, and there is no need for any cofactor for their activity [13]. Lipases have the ability to perform very specific biological and chemical transformations, which has made them incredibly popular in many industries [14]. Lipases have shown their ability to contribute to the tens of billion-dollar market of the bioindustry and are used in the manufacturing of paper (pitch control), biodynamics (lipid exclusion), medicinal products, detergents (cleaning agents), additives of food (enzyme-modifying flavor), cosmetics (lipid exclusion), leather (fat removal from animal skin), and wastewater treatment (decomposition and oil removal). Lipases also find their applications in polymer biodegradation and fatty waste degradation [15]. Lipase is cited as one of the most vital industrial enzymes because of its unique enormous catalytic ability. There are several factors that make lipases excellent biocatalysts. First, they normally show exquisite regioselectivity, chemoselectivity, and stereos electivity. Second, they are easily available in large volumes, and microbial species (fungi and bacteria) can produce them in excess quantities easily. Thirdly, crystal structures of lipases have been generated, which has helped in making the design of logical engineering strategies considerably simpler [16].In the end, they typically do not need cofactors or catalyzed side reactions.

Fungi are considered to be excellent producers of lipase. Most industrially and commercially important fungi that yield lipase belong to *Aspergillus* sp., *Rhizopus* sp., and *Penicillium* [12]. Lipase is used for the fermentation of vegetables, which is beneficial because fatty acid content is increased in fermented vegetables. The fish processing industry achieves a decrease in fat content and increased efficiency through the use of lipase [17]. In the textile industry, lipases are utilized for the extraction of lubricants in the fabric and increase the amount during dyeing. Lipases are also used in the detergent industry, in the paper and pulp industry [18], in pharmaceutical companies for drug production, and also in the cosmetics industry. Physiologically, lipase is important for fat metabolism and has been given as a therapeutic supplement. Lipase was also used to develop quick, effective, and precise biosensors for the qualitative determination of triglycerol and its related substances [19] in different disorders such as cardiovascular dysfunction and screening.

To satisfy the special demands, different hydrolytic enzymes were screened. Fungal lipases were the best source of lipase among the tested microorganisms as they are stable against high temperatures with high turnover numbers and are currently gaining devotion due to their fast retrieval of extracellular enzymes [20]. The production of lipase by different strains of fungi varies depending on the strain and growth medium composition, such as the nitrogen and carbon source, and the temperature [21]. Filamentous fungi are excellent lipase producers among microbial sources, and the extraction, purification, and processing steps are relatively easy. The *Aspergillusaculeatus* fungal strain was extracted from polluted

soil dairy waste and had a lipase activity of 9.51 U/mL. The industrial market for new sources of lipase with different catalytic characteristics promotes the isolation and selection of new fungal strains [22].

On the industrial scale, lipases are produced by using submerged fermentation [23]. High costs of equipment and media and a higher risk of contamination are problems associated with this operation. The conventional and emerging solid-state fermentation (SSF) sector has allowed a large variety of enzymes and metabolites that need less energy than SSF to be better developed especially in developing countries [24]. Solid-state fermentation is now used in newly developed products such as bioactive compounds and organic acids, new trends such as bioethanol and biodiesel as alternative energy sources, and biosurfactant molecules with environmental purposes such as valorizing unexploited biomass [25].

Over the years, SSF technology has been improved and is currently the best way to acquire fungal spores on the insoluble matrix serving as a natural habitat for filamentous fungi. Agricultural residues are produced in enormous amounts in developing countries, but their disposal is also among the main issues as they cause pollution. Using these residues as a source of nutrients for the development of enzymes would therefore reduce the total cost of production. Therefore, during the development of enzymes, suitable substrate selection is an important step. Several agro-industrial leftovers were examined, such as bagasse oil cake, olive oil cake, rice husk, wheat bran, soybean cake, gingelly oil cake, peanut cake, and bagasse sugar cane [26].

Lipases are mainly extracellular and are produced under fermentation conditions such as pH, inoculum size, temperature, carbon sources, and agitation. The main objective of optimizing these parameters is the high production of low-cost fungal lipase by fermentation process. The classical and standard method proved to be less effective and more time-consuming, and more work is needed to obtain optimum conditions, so the best alternative method is used known as the response surface method. It has been an excellent approach in many biotechnological processes [27]. RSM is basically a set of statistical and mathematical techniques used to design experiments and models and find optimal conditions for the parameters under study. For the hyperproduction of fungal lipase, parameters can be effectively optimized by using RSM. Among the response surface designs of RSM, central composite design (CCD) is the most widely used in this design, and the upper and lower limits of each parameter are specified. The continuous demand for industrial lipase production with potential biotechnological applications has necessitated the need to bioprospect native and robust/hyperactive bacterial strains capable of mass production of this enzyme. In this analysis, the optimization of the culture conditions for enhanced lipase production by the native *B. Aryabhattai* SE3-PB has been investigated using RSM. The effect of different inducing oils on the development of lipase has also been studied. For the increased production of lipase, all parameters, such as pH, inoculum volume, temperature, speed of agitation, and moisture content, were optimized by CCD based on face (FCCCD) [28].

After screening, the next step is enzyme purification, which can be performed by the method of Sephadex G-100 gel column chromatography right after the ammonium sulfate precipitation. With maximum purification, lipase will present maximum hydrolytic activity. The objective of this work is to extract and screen various factors through solid-state fermentation, lipase production optimization by response surface methodology, and characterization of lipase from one of the novel local fungal strains.

## 2. Results and Discussion

### 2.1. Strain Identification and Culturing

The novel strain that was obtained for the maximum production of lipase was *Aspergillus niger*. It was examined under a light microscope and identified by its specific growth pattern. The mature fungal surface appear black in color, and these fungi exist in nature as heterotrophs, so they were provided with all the important nutrients required for growth on potato dextrose agar media [29].

## 2.2. Substrate Screening

Eight substrates (rice bran, wheat bran, guava leaves, peanut husk, corn husk, reed grass, sugarcane bagasse) were inoculated with the spores from pure inoculum media to evaluate the best substrate for the production of lipase. The substrate that proved to be best was guava leaves as maximum lipolytic activity was obtained from all other substrates. Previously, an experimental design was used to incorporate sugarcane bagasse, wheat bran, and soybean bran in to the lipase production process by *Aspergillus* sp. and *Penicillium* sp. [30]. Substrates such as apple pomace, beans, corn steep dry, coffee pulp, lemon peel, cane bagasse, and rice husk have also been reported to be used for lipase production [31]. Screening of different substrates for maximum lipase is shown in Figure 1.

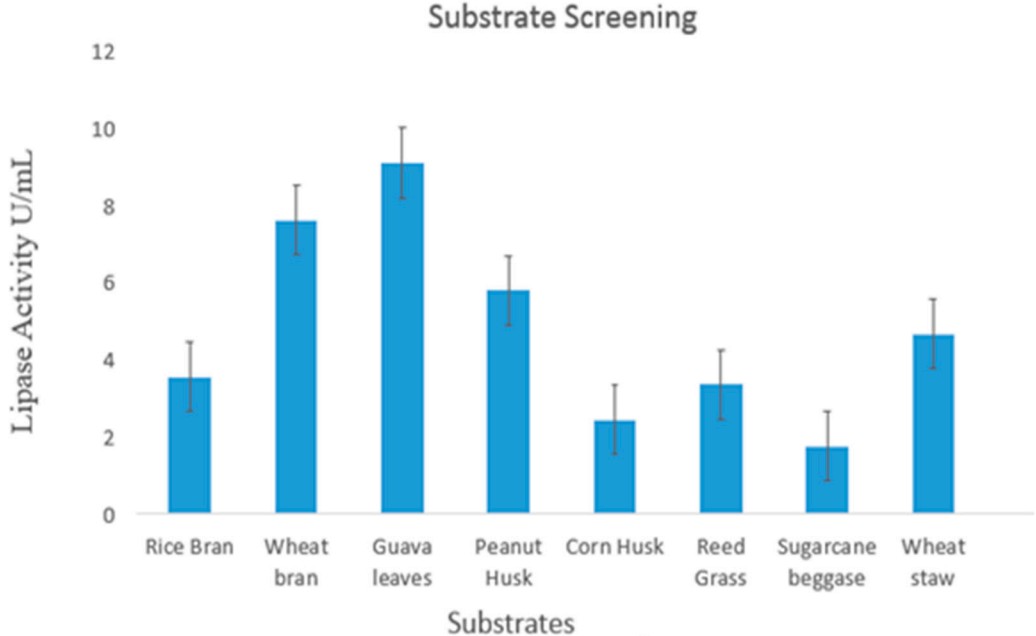

**Figure 1.** Biomass screening of substrates to evaluate the best substrate for Lipase production.

## 2.3. Optimization of Different Physical Parameters through Response Surface Methodology (RSM)

After the screening of substrates having maximum enzyme activity, different physical parameters such as temperature (°C), inoculum size (mL), moisture level (%), incubation period (days), substrate size (g), and pH were optimized by the central composite design (CCD) using response surface methodology for maximum lipase production from *Aspergillus niger* [32].

RSM calculates a response based on a combination of factor levels and determines the best operating conditions to enhance system performance. One of the main advantages of RSM is that it yields a large amount of information from a small number of experiments. The main effects of variables and their interactions on the response can be studied using models and graphical illustrations. It determines the factor levels that provide the best response and the best conditions as a result of multiple responses [33].

The same set of experiments was modeled using two different methods: response surface methodology (RSM) and artificial neural network (ANN). Statistical analyses revealed that both methods, RSM and ANN, can be used to accurately predict response, but the RSM ($R^2$ = 0.9987) method was found to be slightly superior to the ANN model ($R^2$ = 0.9973).

RSM trials for the optimization of physical parameters to obtain the maximum lipase activity (U/mL) from *Aspergillus niger* is shown in Table 1.

**Table 1.** Optimization via RSM trials along with lipase activity U/mL.

| Sr # | Substrate Size (g) | pH | Temperature (°C) | Inoculum Size (mL) | Moisture Level (%) | Incubation Time (Days) | Activity U/mL |
|---|---|---|---|---|---|---|---|
| 1 | 4 | 2 | 45 | 5 | 60 | 5 | 4.202 |
| 2 | 4 | 2 | 37 | 5 | 60 | 3 | 4.828 |
| 3 | 2 | 2 | 40 | 2 | 100 | 4 | 8.212 |
| 4 | 2 | 2 | 45 | 3 | 20% | 6 | 5.691 |
| 5 | 6 | 5.5 | 45 | 3 | 100 | 3 | 7.975 |
| 6 | 8 | 5.5 | 40 | 2 | 40 | 3 | 7.061 |
| **7** | 4 | 5.5 | 40 | 4 | 60 | 4 | 8.922 |
| 8 | 6 | 5.5 | 37 | 2 | 100 | 8 | 8.347 |
| 9 | 8 | 8 | 40 | 2 | 20 | 8 | 11.76 |
| 10 | **6** | **8** | **40** | **3** | **40** | **3** | **12.52** |
| 11 | 4 | 8 | 45 | 2 | 60 | 6 | 8.872 |
| 12 | 2 | 8 | 45 | 1 | 100 | 8 | 7.501 |
| 13 | 4 | 9.5 | 30 | 8 | 60 | 1 | 7.823 |
| 14 | 4 | 9.5 | 30 | 3 | 125 | 4 | 9.413 |
| 15 | 4 | 9.5 | 40 | 1 | 60 | 6 | 7.738 |
| 16 | 7 | 9.5 | 50 | 4 | 60 | 8 | 7.146 |
| 17 | 4 | 11 | 30 | 5 | 50 | 1 | 4.625 |
| 18 | 4 | 11 | 30 | 5 | 60 | 3 | 6.080 |
| 19 | 1 | 11 | 52 | 3 | 60 | 4 | 5.640 |
| 20 | 4 | 11 | 45 | 3 | 60 | 8 | 7.603 |

### 2.3.1. Temperature Optimization of Lipase

For the production of enzymes from fungi, temperature is an important factor, and different trials of temperatures ranging from (30–52°C) were illustrated by RSM. Results obtained from RSM showed that the maximum activity of lipase was at 40°C in the solid-state fermentation media, and thus this temperature proved to be the optimum temperature for this fungal strain to produce lipase. From different studies, it was shown that fungi show maximum growth in the temperature range of 35–45°C. When the temperature is above or below 40°C, the metabolic activity of fungi slows down, eventually resulting in less enzyme production [34]. Temperature optimization of lipase activity is shown in Figure 2.

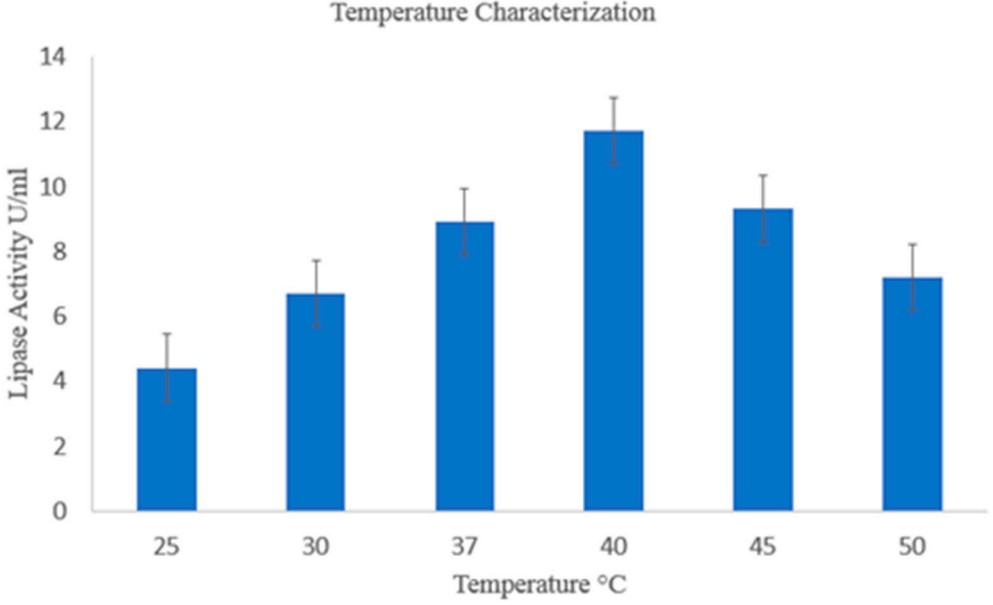

**Figure 2.** Temperature optimization of lipase activity.

### 2.3.2. pH Optimization of Lipase

Buffers with different pH values (2–9.5) that were also a source of moisture were added to the growth media to their respective trials according to RSM design for the determination of optimum pH for lipase. Guava leaves favored the growth of fungi at all pH sets, but the maximum lipase activity was at pH 8. Fungal lipase is alkaline in nature; it was proved by the results too. Optimization of pH is a critical parameter because fungal growth is affected by it as it helps in the binding of fungi to substrate, transfer of nutritional components, and formation of products. Many studies support the evidence that lipase produced from Aspergillus gives maximum growth in the pH range of 8–10 [35].

### 2.3.3. Optimization of Incubation Period for Lipase

For the cultivation of fungi in the fermented media, time period is a crucial factor; the harvesting time period was optimized by RSM to obtain maximum enzyme activity. Fungi were given a growth period of 1 to 6 days to make their interaction with the substrate, and then an enzyme assay was performed to determine the best growth period. The activity of fungi for lipase production was evaluated at regular intervals of 24 h [28].In nitrogen-rich media, maximum lipase activity was recorded after 96 h. When the growth period was decreased or increased, it resulted in the release of fetal metabolites and eventually improper growth [36].

### 2.3.4. Inoculum Size Optimization for Lipase Production

Inoculum size is also considered in RSM design to provide support for fungal cultivation. Inoculum size varying from 1 to 7 mL was used in the trials for lipase extraction. Over the substrate surface, fungal spores are uniformly distributed; they penetrate into the substrate for the good production of enzymes. Within the set incubation time, the inoculum size that showed the maximum lipase activity was 3 mL. It may not seem sufficient that a smaller inoculum size will favor the maximum enzyme activity, but when favorable conditions are present, such as time period, temperature, and pH, these spores are enough to give maximum activity. Many other citations are also in favor of smaller inoculum size because, with the increase in inoculum size, negative impacts such as depletion of nutrients and oxygen were observed [37].

### 2.3.5. Optimization of Substrate Concentration for Lipase

Among all the factors, substrate concentration was highly encouraged in RSM trials, ranging from 1 to 8 g for the production of lipase. Guava leaves with different substrate concentrations were taken into flasks and then inoculated with fungi and different moisture contents as optimized by RSM trials. Then, the flasks were placed in incubators at different temperature sets, guided by RSM, to determine the optimum substrate concentration for the production of lipase at different parameter sets. After harvesting the media, it was filtered and then centrifuged for enzyme assay using a mass spectrophotometer. Results taken from the absorbance at the spectrophotometer reported that maximum lipase activity was observed with 6 g of guava leaves when they were given with an optimum pH of 8, 40 °C temperature, and an incubation period of 4 days. This lipase activity was observed after standardizing the substrate. When the substrate concentration was increased, it was observed that growth rate was decreased because of the smaller diffusion rate of fungi into substrate spaces [38].

### 2.4. Statistical Graphs for Lipase (3D and Contour)

To design the statistical graphs, Design Expert software was used. Two types of graphs were made: one was a three-dimensional response surface graph, and the other was a quadratic contour plot. The relationship among the applied parameters, such as temperature, inoculum size, pH, and substrate size, in accordance with the activity of lipase, was shown by response surface graphs and their 3D structures.

### 2.4.1. Interaction of pH and Incubation Period for Lipase Activity

A response surface plot is a 3D plot that shows the relationship between independent variables and response. A contour plot is a 2D surface plot display, and here, constant response lines were drawn at the level of independent variables. Response surface shape was visualized by the contour plot. Ellipses or the center of circles shown by the contour plot were the points of minimum and maximum response. Elliptical contours showed the maximum interaction of variables [39].

To obtain the maximum lipase activity, our plot was constructed between the incubation period and pH. The contour plot of pH versus time period showed that the production of lipase was increased when we used buffers in the range of pH 7 to 9.5, and the maximum lipase activity was seen at pH 8. Contour plot of enzyme activity U/mL vs. pH and incubation period is shown in Figure 3.

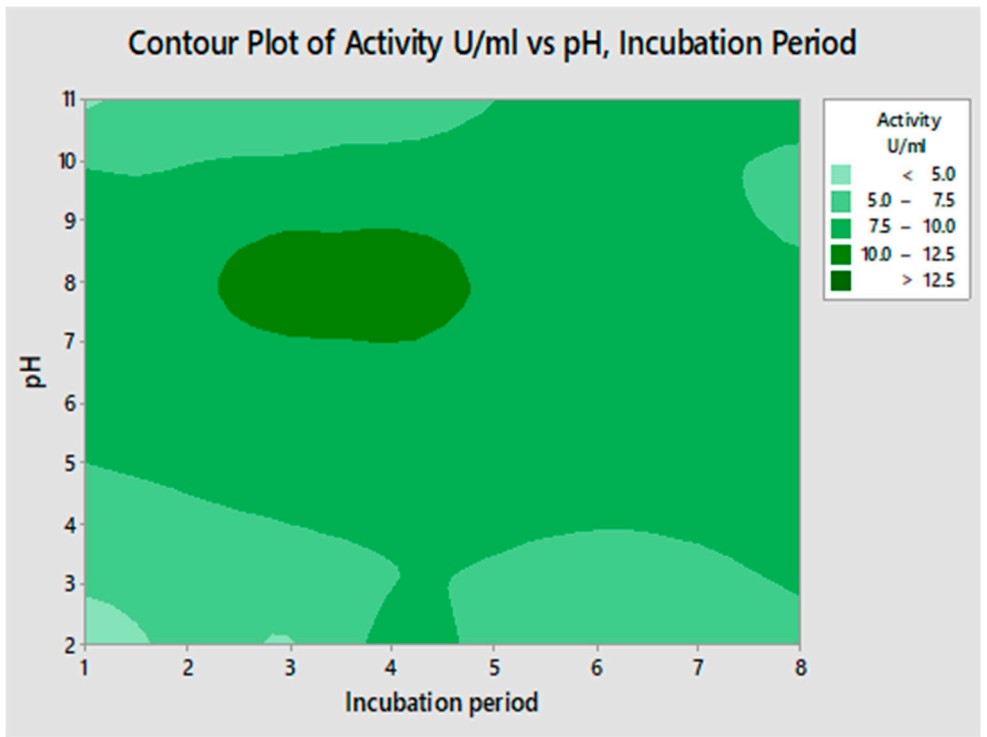

**Figure 3.** Contour plot of enzyme activity U/mL vs. pH and incubation period.

Similarly, increasing lipase activity was seen after 2 days of incubation and continued until the 5th day, after which lipase activity was significantly reduced. Maximum lipase activity was on the 4th day of incubation. Therefore, concerning time period, it was seen that at pH 8, it showed maximum lipase activity. The figure shows the 3D surface plot between incubation period and pH. In this plot, enzyme activity U/mL is shown on the *z*-axis, and its interaction is shown with incubation period on the *y*-axis and pH on the *y*-axis. The peaks in this plot indicate maximum enzyme activity at an incubation period of 4 days against pH 8 (shown on the *y*-axis). Surface plot of enzyme activity U/mL vs. pH and incubation period is shown in Figure 4. Results from the 3D surface plot support the results obtained from the contour plots as more activity was observed at alkaline pH after 4 days of incubation.

### 2.4.2. Interaction of pH and Temperature for Lipase Activity

To obtain the maximum lipase activity, a contour plot was constructed between temperature and pH. The contour plot of pH versus time period showed that the production of lipase was increased when we used a buffer of pH 8. Similarly, the increasing lipase activity was obtained at the temperature of 40°C. Therefore, concerning temperature, it was seen

that at pH 8, it showed maximum activity of lipase. This plot also indicated that for the production of lipase, we need a temperature in the range of 37–43°C. because in this plot, the enzyme activity was increasing at 37°C, and the best yield was obtained at 40°C. The contour plot of enzyme activity U/mL vs. pH and temperature is shown in Figure 5. 3D surface plot between temperature and pH is also constructed in this plot, enzyme activity U/mL is shown on the z-axis, and its interaction is shown with temperature on the y-axis and pH on the *y*-axis. The peaks in this plot indicate maximum enzyme activity at pH 8 and 40°C temperature. Surface plot of enzyme activity U/mL vs. pH and temperature is shown in Figure 6 Results from the 3D surface plot support the results obtained from the contour plots as more activity is observed at alkaline pH and high temperature.

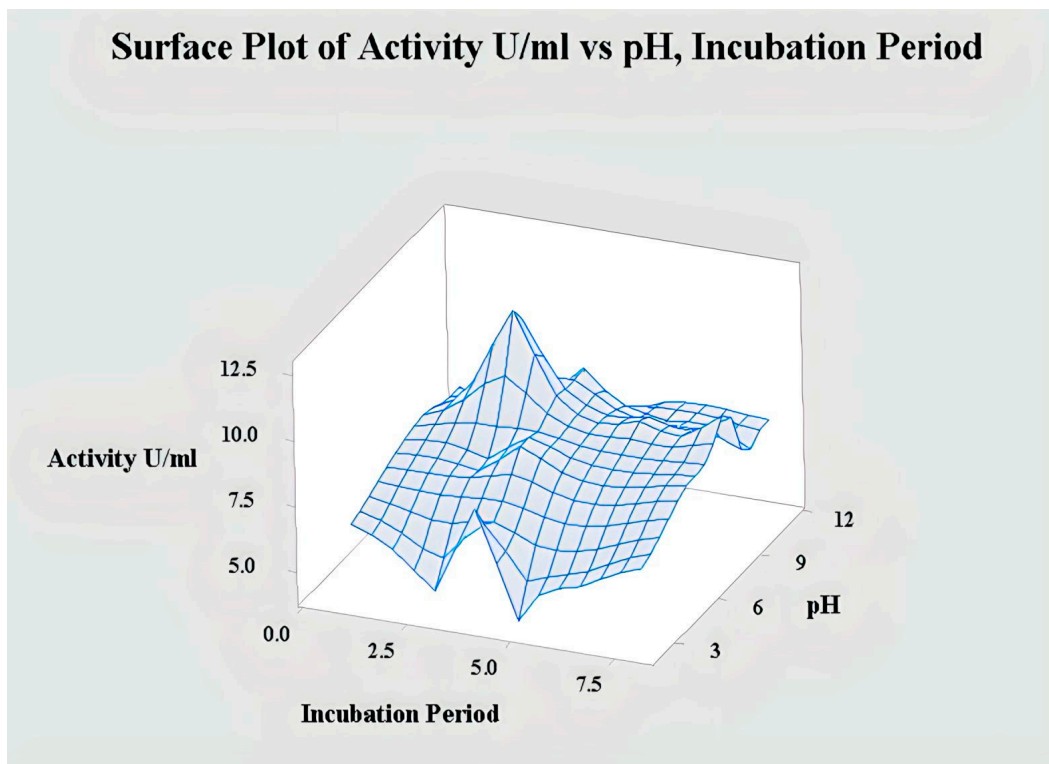

**Figure 4.** Surface plot of enzyme activity U/mL vs. pH and incubation period.

### 2.4.3. Interaction of pH and Substrate Concentration for Lipase

To obtain the maximum lipase activity, a contour plot was constructed between substrate concentration and pH. Higher production of lipase was obtained when pH was in the range of 7–9.5 and the substrate concentration was 6 to 8 g. Maximum activity was observed at a substrate concentration of 6g and 8 pH. Any further increase or decrease from this incubation period and pH resulted in a decrease in lipase activity. Contour plot of enzyme activity u/mL substrate concentration and pH is shown in Figure 7.

Enzyme activity U/mL indicated by the 3D surface plot also supported the results obtained from the contour plots. Peaks on the 3D surface indicate that enzyme activity was highest with a 6g substrate concentration (shown on the *y*-axis) and at 8 pH (shown on the *x*-axis). Results from the 3D surface plot support the results obtained from the contour plot as more activity is observed. Results from the 3D surface plot support the results obtained from the contour plots, as when substrate values are higher and pH values are lower, fungal growth is prohibited because its metabolic rate slowed down, and hence less lipase activity was observed [40]. The surface plot of enzyme activity u/mL substrate concentration and pH is shown in Figure 8.

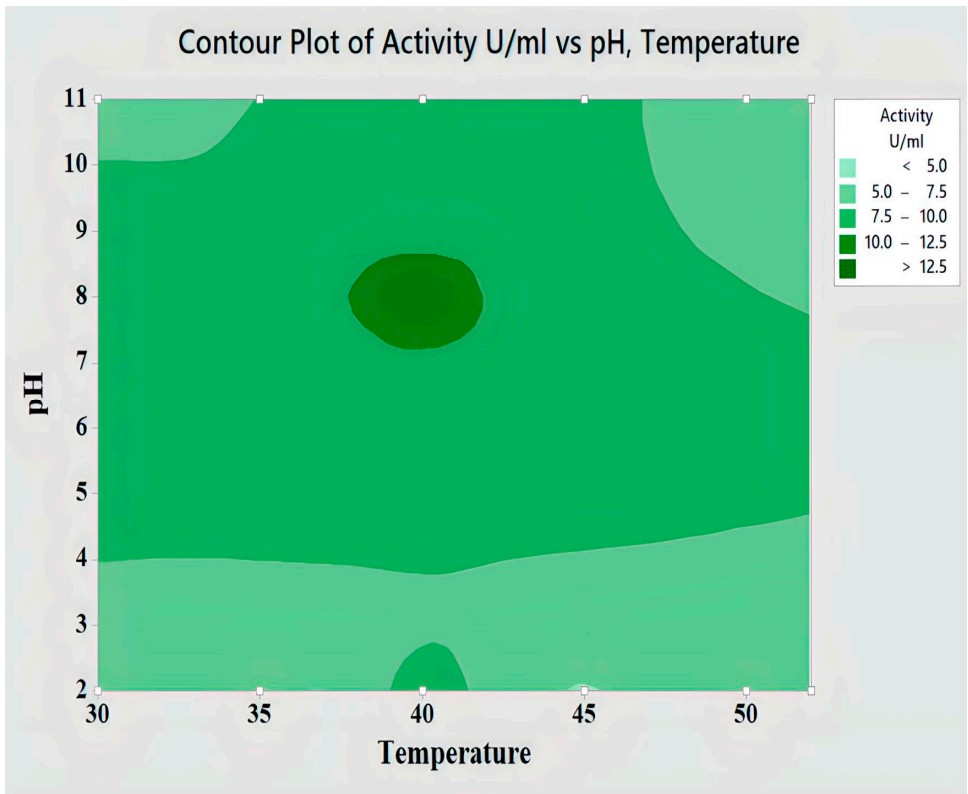

**Figure 5.** Contour plot of enzyme activity U/mL vs. pH and temperature.

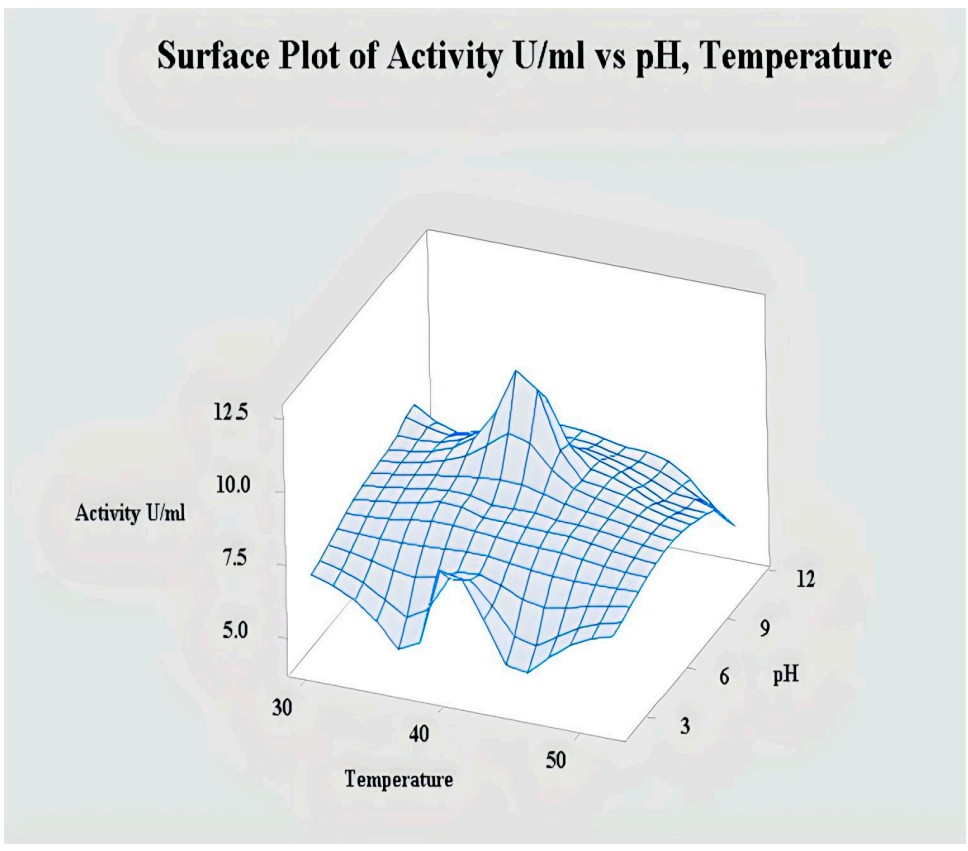

**Figure 6.** Surface plot of enzyme activity U/mL vs. pH and temperature.

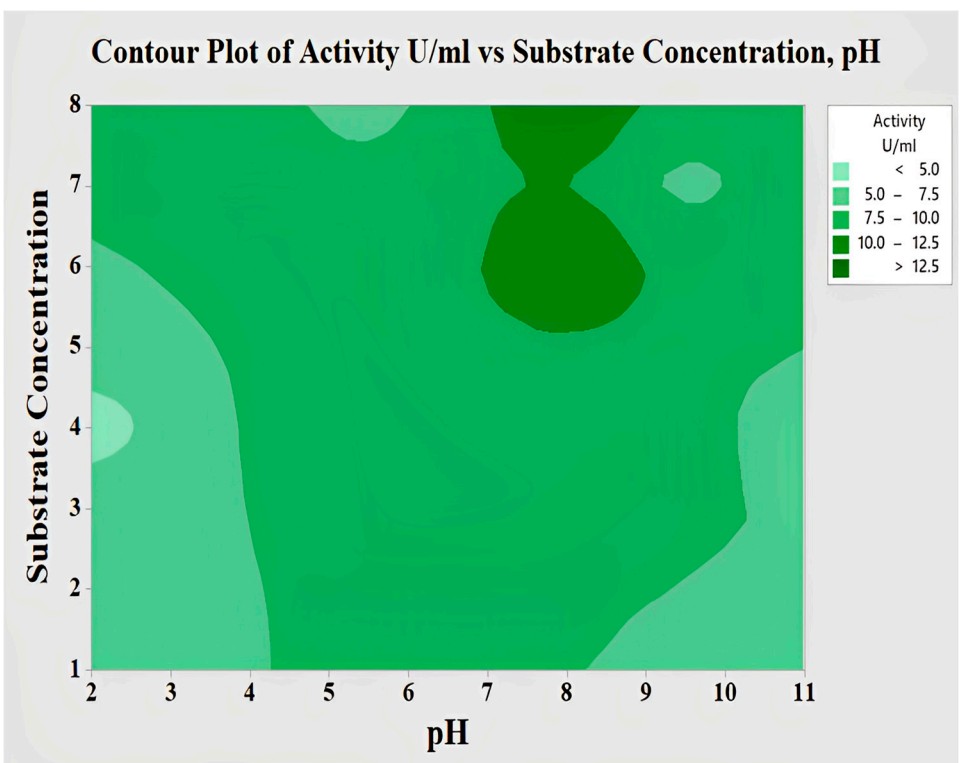

**Figure 7.** Contour plot of enzyme activity u/mL substrate concentration and pH.

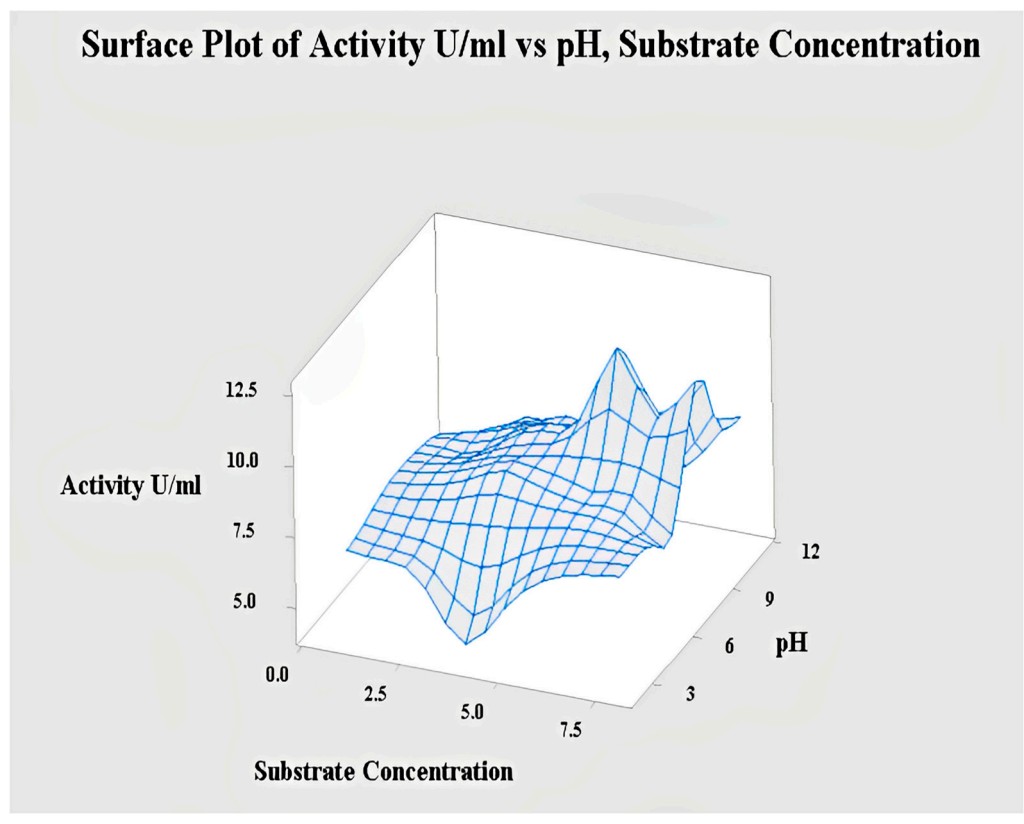

**Figure 8.** Surface plot of enzyme activity u/mL substrate concentration and pH.

### 2.4.4. Interaction of pH and Inoculum Size for Lipase

To obtain the maximum lipase activity, a contour plot was constructed between pH and inoculum size. Lipase production was increased when the pH was between 7 and 8.5 and the inoculum size was 2–4 mL. Maximum activity was observed at an inoculum size of 3mL, as shown on the *y*-axis, and a pH of 8. Any further increase or decrease in this inoculum size and pH resulted in a decrease in lipase activity. The contour plot of enzyme activity u/mL vs. pH and inoculum size is shown in Figure 9.

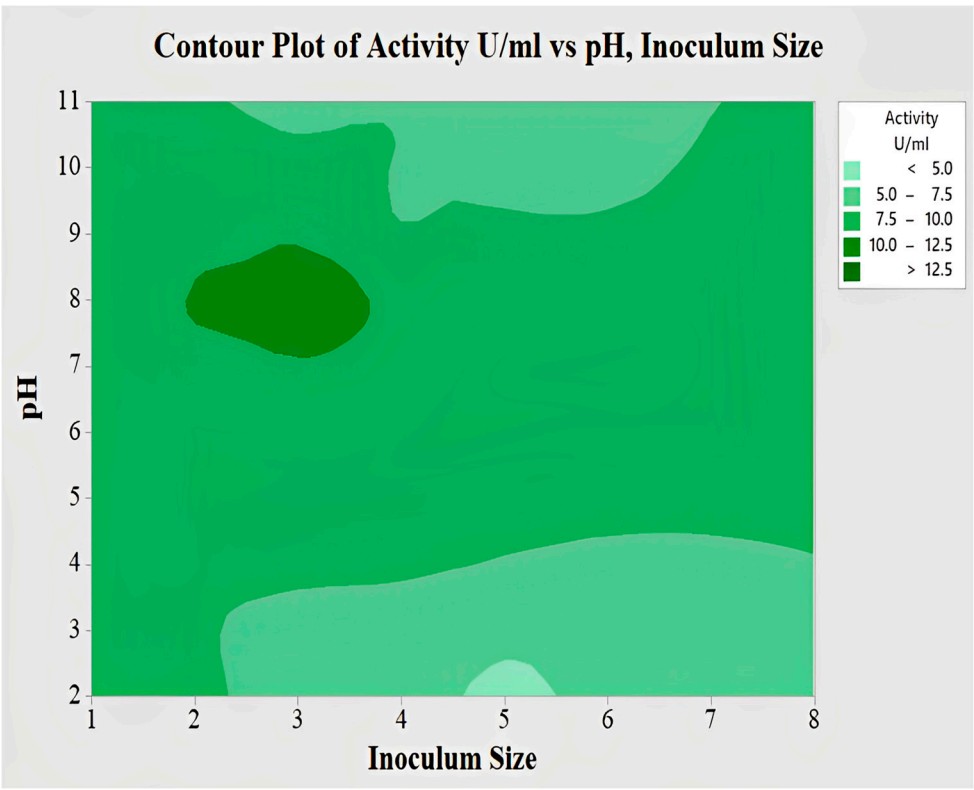

**Figure 9.** Contour plot of enzyme activity u/mL vs. pH and inoculum size.

Enzyme activity U/mL indicated by the 3D surface plot supports the results obtained from the contour plots. Peaks of the 3D surface indicate that enzyme activity was highest with an inoculum size of 3mL (shown on the *x*-axis) and at 8 pH (shown on the *y*-axis). Fungal growth has negative impacts when inoculum size is higher because of problems such as less space and nutrient depletion [41]. The surface plot of enzyme activity u/mL vs. pH and inoculum size is shown in Figure 10.

### 2.4.5. Interaction of pH and Moisture Content for Lipase Activity

To obtain the maximum lipase activity, a contour plot was constructed between pH and moisture content. The contour plot of pH versus moisture is shown in Figure 11, it illustrates that the production of lipase increased when we used a buffer of pH 8. Similarly, the increase in lipase activity was obtained with a moisture content of 20–45%. Concerning the moisture content, it was seen that at pH 8, it showed maximum activity of lipase. This plot also indicated that for the production of lipase, we need moisture content from 20 to 45%. The 3D surface plot between moisture content and pH is shown in Figure 12. In this plot, enzyme activity (U/mL) is shown on the *z*-axis and its interaction with moisture on the *x*-axis and pH on the *y*-axis. The peaks in this plot indicate maximum enzyme activity. Therefore, more enzyme was produced at an alkaline pH and moderate moisture level.

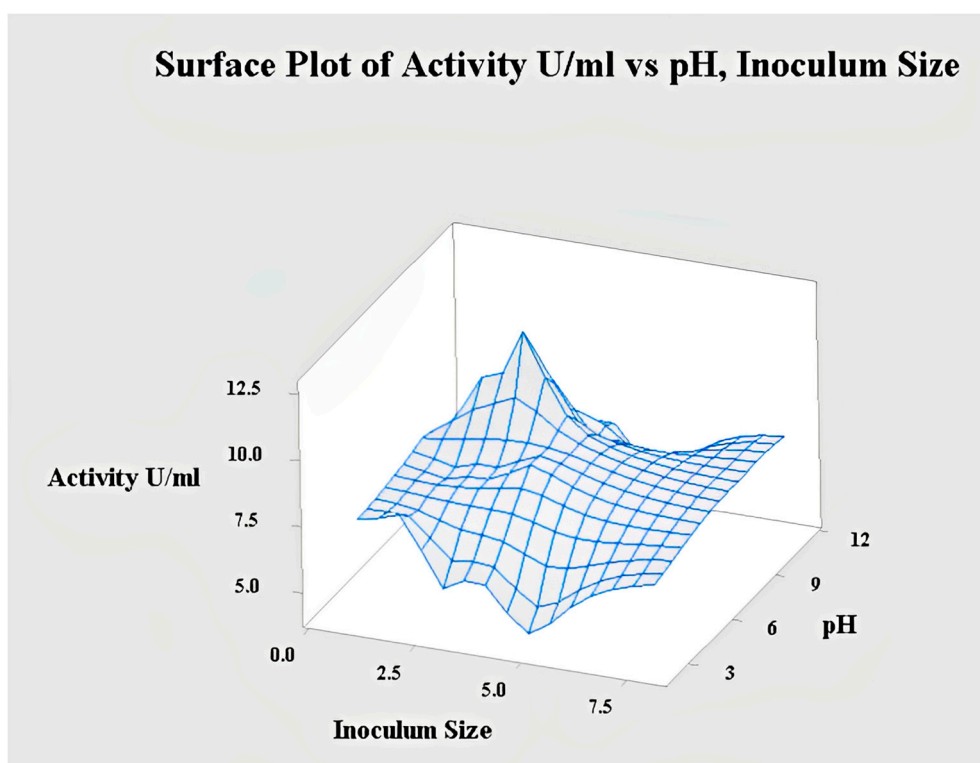

**Figure 10.** Surface plot of enzyme activity u/mL vs. pH and incubation size.

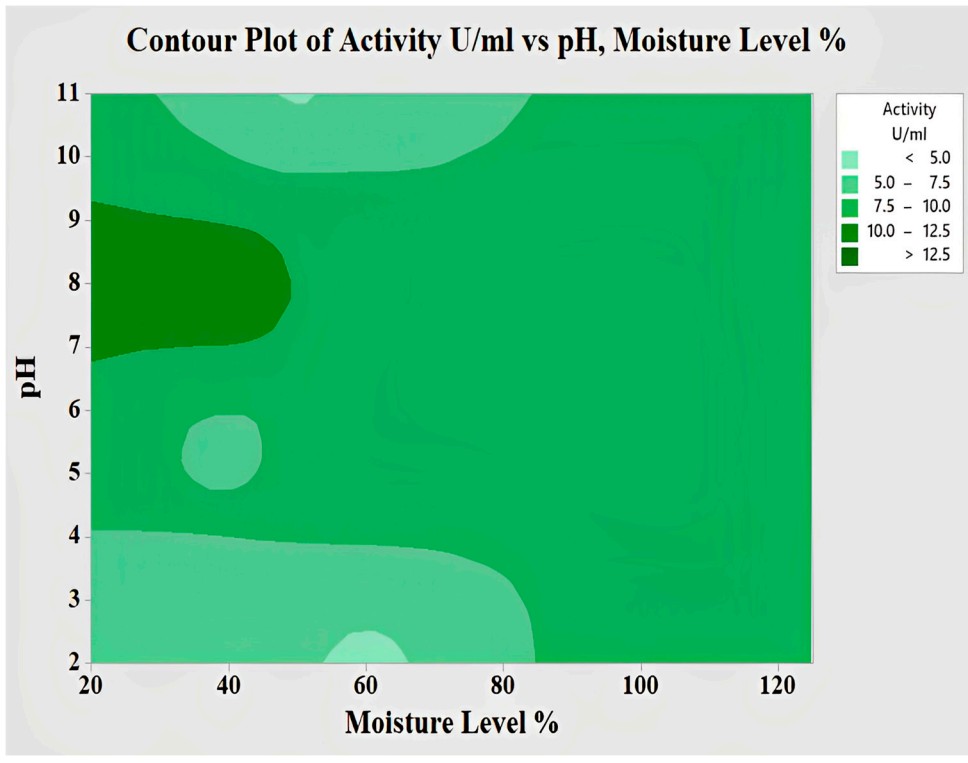

**Figure 11.** Contour plot of enzyme activity u/mL vs. pH and moisture level.

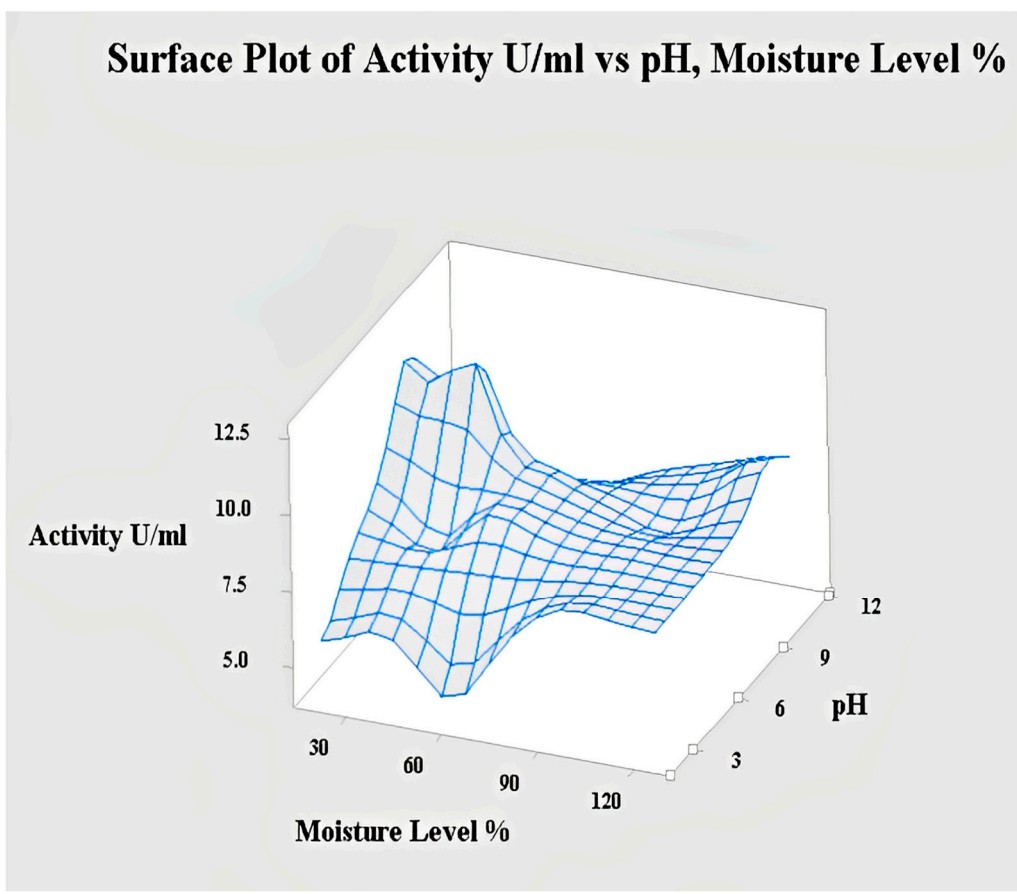

**Figure 12.** Surface plot of enzyme activity u/mL vs. pH and moisture level.

2.4.6. Interaction of Inoculum Size and Incubation Period for Lipase

To obtain the maximum lipase activity, a contour plot was constructed between inoculum size and incubation period. Maximum activity was observed at an incubation period of 4 days, as shown on the *x*-axis, and an inoculum size of 3mL. Any further increase or decrease from this incubation period and inoculum size resulted in a decrease in lipase activity. Interaction of inoculum size and incubation period for lipase is shown by contour plot in Figure 13.

Optimum enzyme activity indicated by the 3D surface plot supported the results obtained from the contour plots. Peaks on the 3D surface indicate that enzyme activity was highest at an incubation period of 4 days, as shown on the *x*-axis, and the inoculum size of 3 mL, as shown on the *y*-axis. The surface plot for the interaction of inoculum size and incubation period for lipase is shown in Figure 14.

2.4.7. Interaction of Inoculum Size and Moisture Level for Lipase Activity

To obtain the maximum lipase activity, a contour plot was constructed between inoculum size and moisture level, its shown in Figure 15. Maximum lipase activity was obtained when inoculum size was 2 to 3mL with a moisture content of 20% and 40%, respectively. Any further increase or decrease from this incubation period and moisture level will result in a decrease in lipase activity.

Enzyme activity U/mL indicated by the 3D surface plot shown in Figure 16, supported the results obtained from the contour plots. When inoculum size was increased, fungi competed with one another for nutrients and space, which resulted in less enzyme activity.

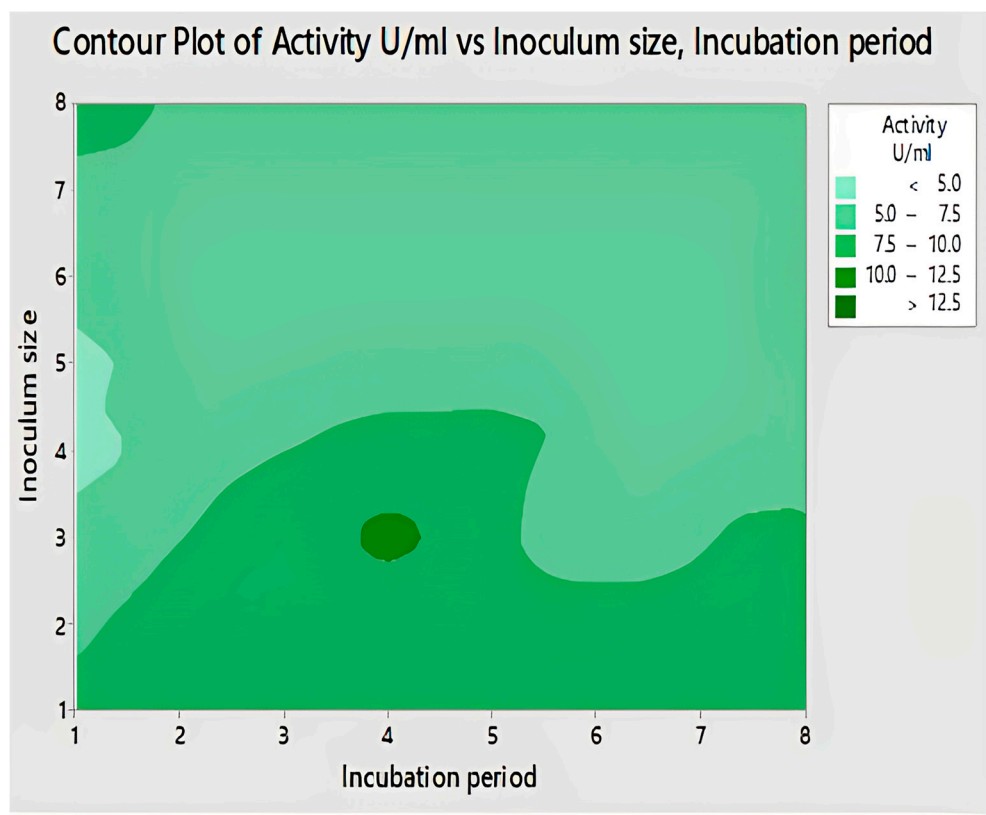

**Figure 13.** Contour plot of enzyme activity u/mL vs. inoculum size and incubation period.

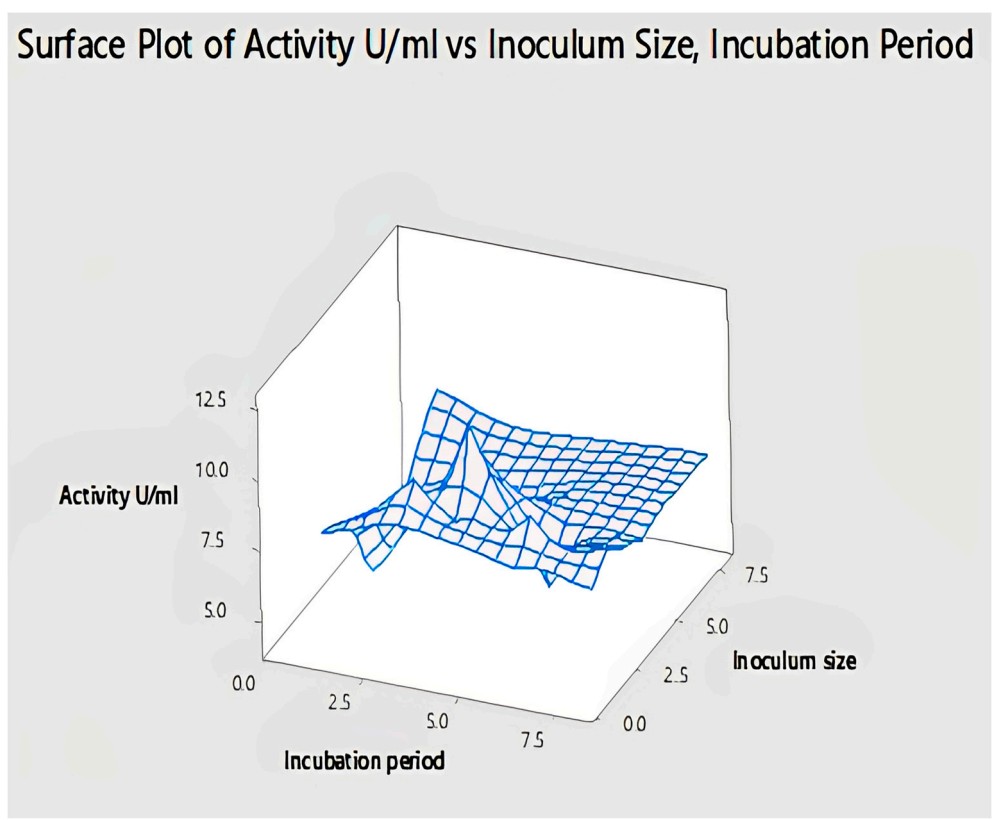

**Figure 14.** Surface plot of enzyme activity u/mL vs. inoculum size and incubation period.

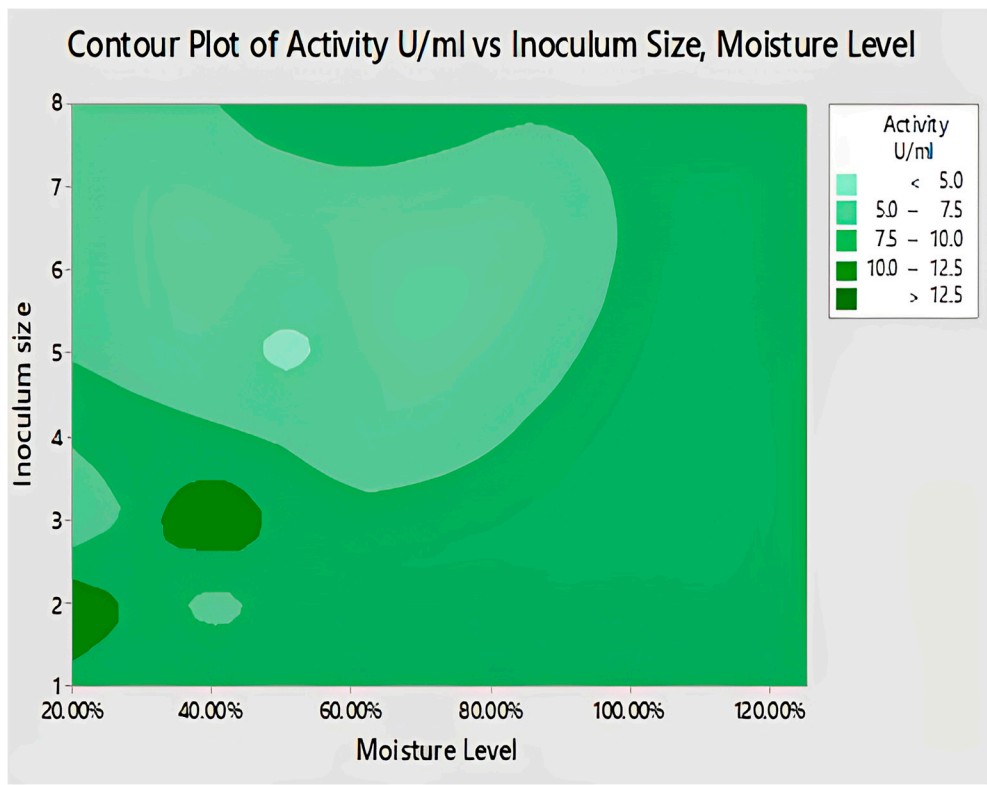

**Figure 15.** Contour plot of enzyme activity U/mL vs. inoculum size and moisture level.

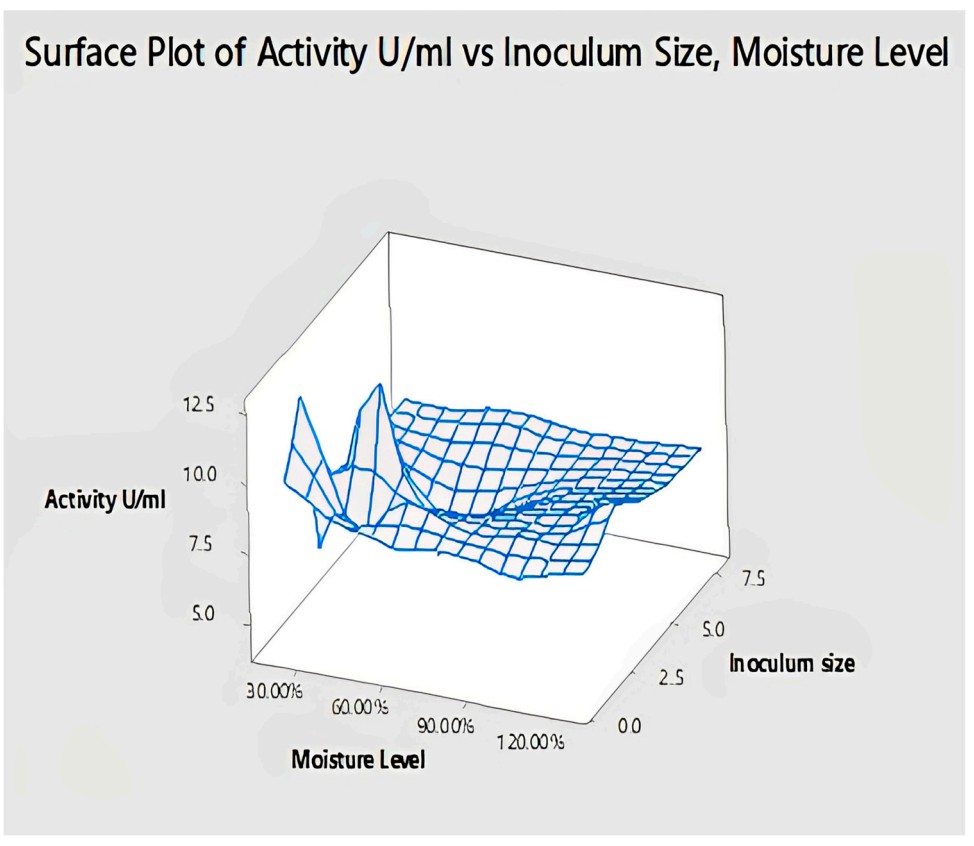

**Figure 16.** Surface plot of enzyme activity U/mL vs. inoculum size and moisture level.

### 2.4.8. Interaction of Temperature and Inoculum Size for Lipase

To obtain the maximum lipase activity, a contour plot was constructed between temperature and inoculum size, shown in Figure 17. Higher production of lipase was observed when the temperature was in the range of 35–43 °C against an inoculum size of 2.5–3.5 mL. Furthermore, 40 °C was the optimal temperature at which lipase activity was at its maximum with an inoculum size of 3mL. Enzyme activity indicated by the 3D surface plot shown in Figure 18, also supported the results obtained from the contour plots. Negative impacts on fungal strains were reported when temperature was further increased and inoculum size was large.

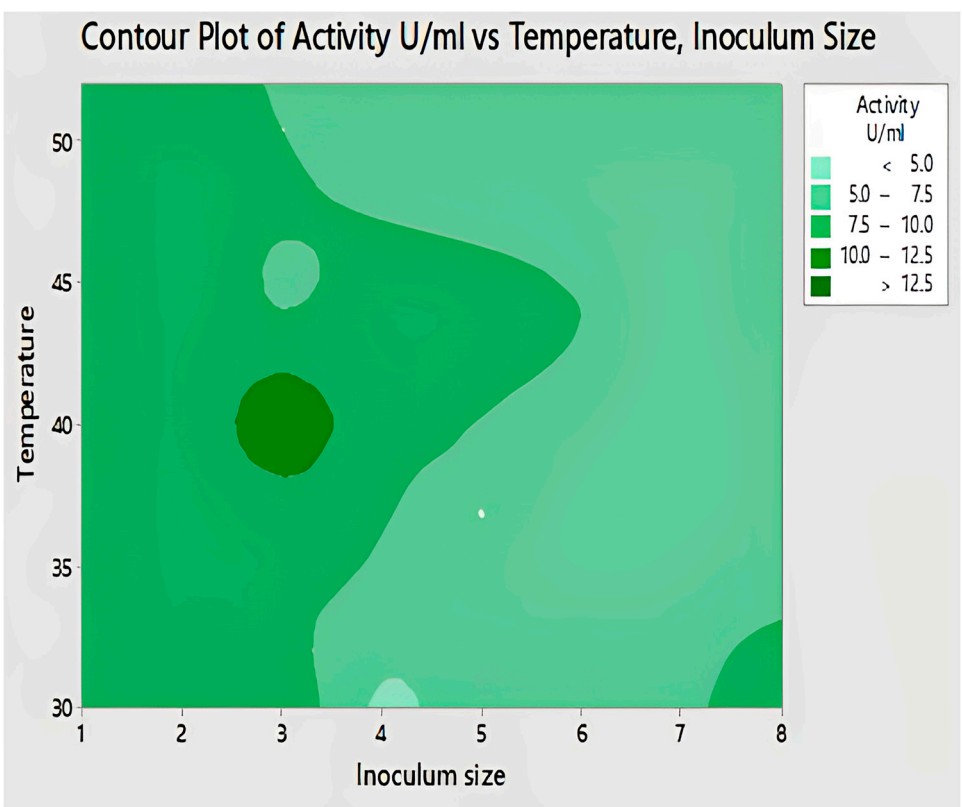

**Figure 17.** Contour plot of enzyme activity U/mL vs. temperature and inoculum size.

### 2.4.9. Interaction of Temperature and Moisture Level for Lipase Activity

To obtain the maximum lipase activity, a contour plot was constructed between temperature and moisture level shown in Figure 19 Increasing lipase activity was obtained at an incubation period of 4 days. Therefore, concerning temperature, it was seen that a moisture level of 20% showed maximum activity of lipase. This plot also indicated that for the production of lipase, we need a temperature in the range of 37–43 °C because in this plot, the enzyme activity was increasing at 37 °C, and the best yield was obtained at 40 °C. The results obtained from the contour plot were also supported by 3D surface plots as shown in Figure 20. In this plot, enzyme activity U/mL is shown on the z-axis, and its interaction with temperature on the y-axis and moisture level on the x-axis is shown. The peaks in this plot (3D surface plot) indicate maximum enzyme activity. Therefore, more enzyme activity was observed at high temperature and moderate moisture range.

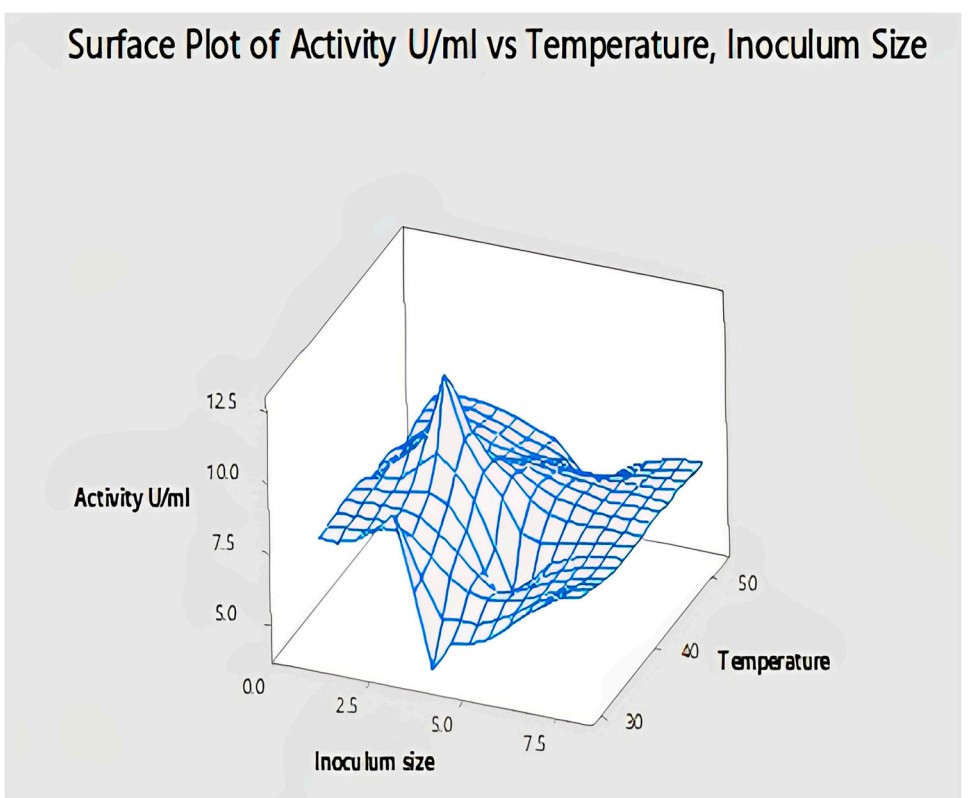

**Figure 18.** Surface plot of enzyme activity U/mL vs. temperature and inoculum size.

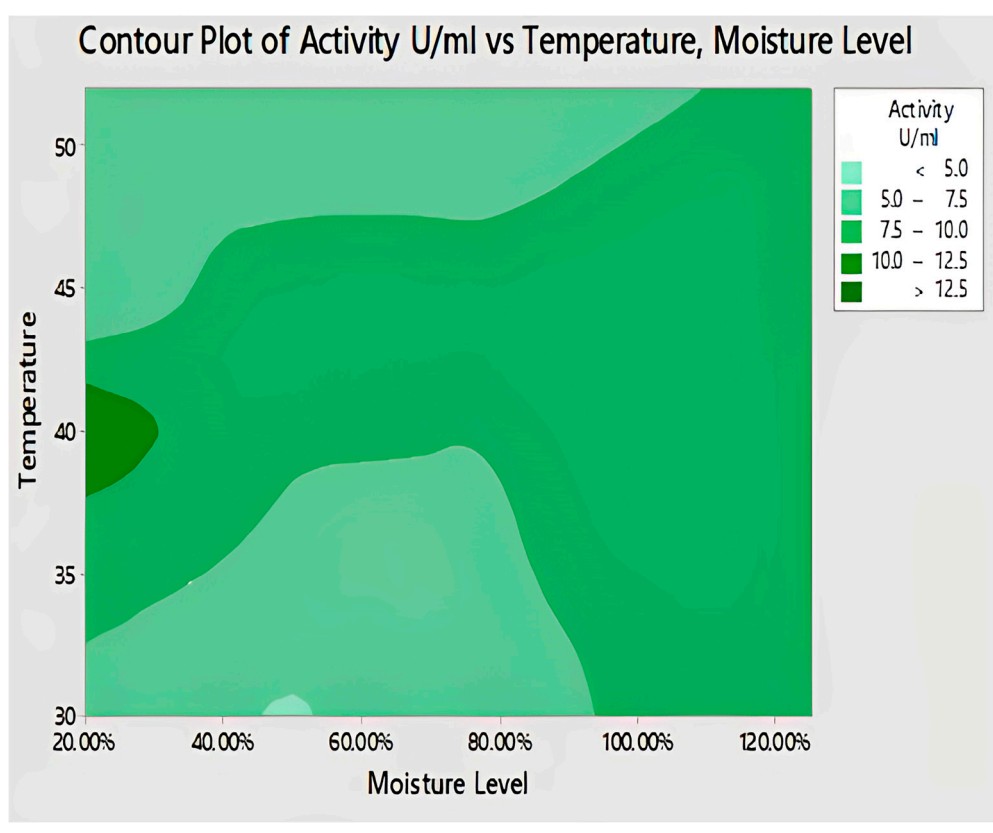

**Figure 19.** Contour plot of enzyme activity U/mL vs. temperature and moisture level.

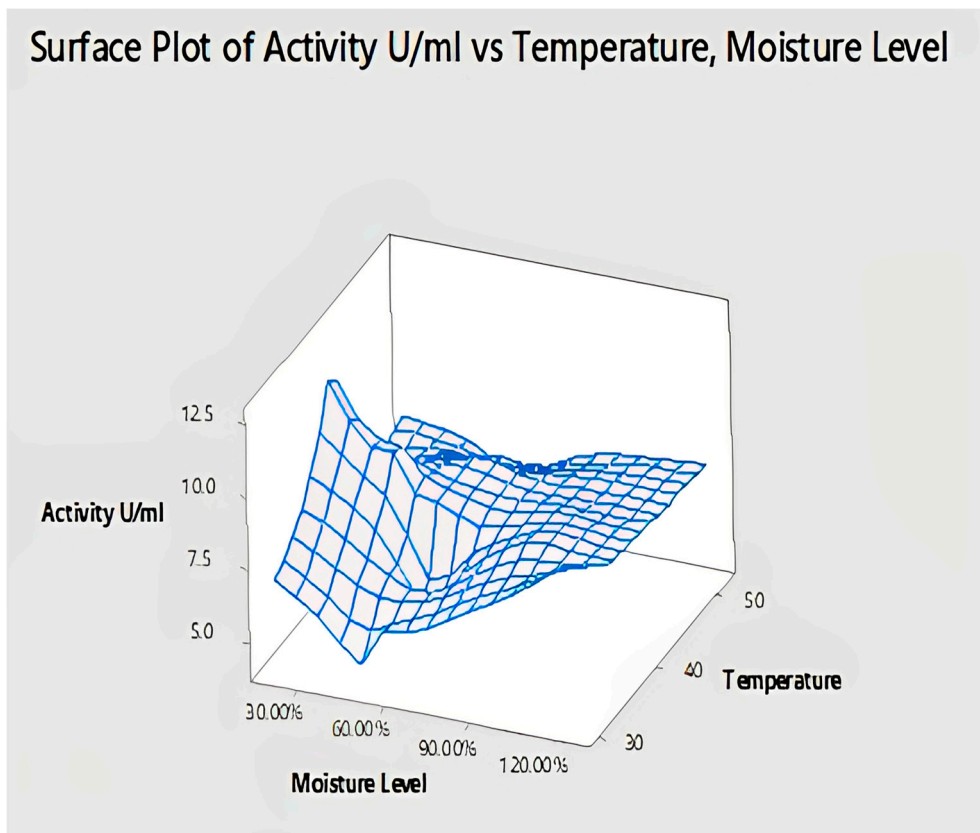

**Figure 20.** Surface plot of enzyme activity U/mL vs. temperature and moisture level.

2.4.10. Interaction of Temperature and Incubation Period for Lipase Activity

To obtain the maximum lipase activity, a contour plot was constructed between temperature and incubation period. The contour plot of pH versus time period is shown in Figure 21, it showed that the production of lipase increased on the 4th day. Therefore, concerning the time period, it was seen that at a temperature of 40°C, it showed maximum activity of lipase. This plot also indicated that for the production of lipase, we need a temperature in the range of 40–50°C because in this plot, the enzyme activity was increasing at 37°C, and the best yield was obtained at 40°C. The Figure 22 shows the 3D surface plot between temperature and incubation period. In this plot, enzyme activity U/mL is shown on the *z*-axis, and its interaction with temperature on the *y*-axis and incubation period on the *x*-axis is shown. The peaks in this plot indicate maximum enzyme activity. Therefore, more enzyme was produced after an incubation period of 4 days and high temperature.

*2.5. Interpretations of Regression Surface Analysis*

Three-dimensional (3D) surface plots and contour plots represent that substrate size, pH, temperature, moisture level, and incubation days showed maximum contribution to the production of lipase. The $R^2$ value is 100%, which shows the model fitness. The following equation can be used to determine the production of lipase.

$$\text{Activity U/mL} = -65.53 + 13.84 \text{ pH}$$

$$-1.724 \text{ Temperature} + 10.37 \text{ Inoculum size}$$

$$+111.6 \text{ Moisture Level} + 4.270 \text{ Incubation period}$$

$$-0.3793 \text{ pH*} \times \text{pH}$$

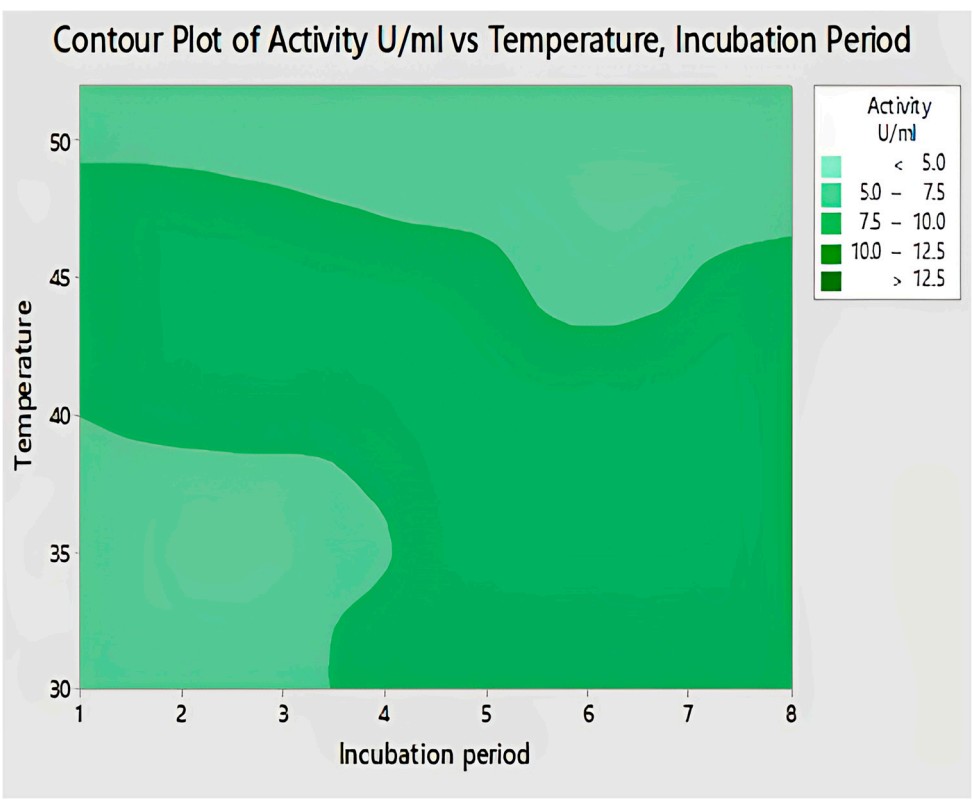

**Figure 21.** Contour plot of enzyme activity U/mL vs. temperature and incubation period.

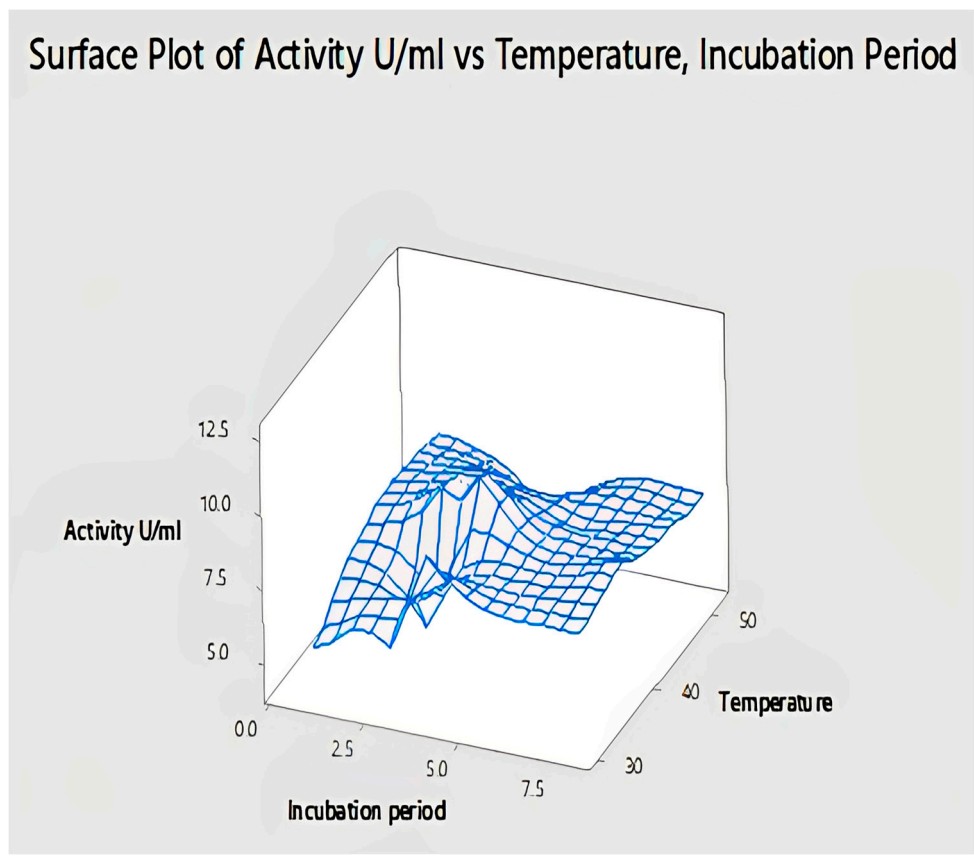

**Figure 22.** Surface plot of enzyme activity U/mL vs. temperature and incubation period.

$$+0.06607 \text{ Temperature} \times \text{Temperature}$$

$$-0.4737 \text{ Inoculum size} \times \text{Inoculum size}$$

$$+16.61 \text{ Moisture Level} \times \text{Moisture Level}$$

$$-0.08778 \text{ Incubation period} \times \text{Incubation period}$$

$$-0.2374 \text{ pH} \times \text{Temperature}$$

$$-0.03243 \text{ pH} \times \text{Inoculum size}$$

$$-5.456 \text{ pH} \times \text{Moisture Level}$$

$$+0.8954 \text{ pH} \times \text{Incubation period}$$

$$+0.1320 \text{ Temperature} \times \text{Inoculum size}$$

$$-2.153 \text{ Temperature} \times \text{Moisture Level}$$

$$-0.1477 \text{ Temperature} \times \text{Incubation period}$$

$$-9.772 \text{ Inoculum size} \times \text{Moisture Level}$$

$$-1.286 \text{ Inoculum size} \times \text{Incubation period}$$

when two-way interaction was studied between different parameters, then all the interactions between different parameters showed significant values because the R-square value is 1, which indicates the high correlation between these parameters for the production of lipase. Regression surface analysis: enzyme activity versus substrate size, pH, temperature, inoculum size, moisture level, and incubation days is shown in Table 2.

**Table 2.** Regression surface analysis: enzyme activity versus substrate size, pH, temperature, inoculum size, moisture level, and incubation days.

| Analysis of Variance | | | | | |
|---|---|---|---|---|---|
| **Source** | **DF** | **Adj SS** | **Adj MS** | **F-Value** | ***p*-Value** |
| **Model** | 19 | 87.801 | 4.62111 | 0.000 | 0.000 |
| **Linear** | 5 | 4.6304 | 0.92608 | 0.000 | 0.000 |
| Ph | 1 | 0.9406 | 0.94063 | 0.000 | 0.000 |
| Temperature | 1 | 1.3550 | 1.35496 | 0.000 | 0.000 |
| Inoculum Size | 1 | 1.3483 | 1.34826 | 0.000 | 0.000 |
| Moisture Level | 1 | 1.1967 | 1.19666 | 0.000 | 0.000 |
| Incubation Days | 1 | 2.2672 | 2.26723 | 0.000 | 0.000 |
| **Square** | 5 | 26.0522 | 5.21044 | 0.000 | 0.000 |
| pH × Ph | 1 | 6.6295 | 6.82954 | 0.000 | 0.000 |
| Temperature × Temperature | 1 | 0.9211 | 0.92106 | 0.000 | 0.000 |
| Inoculum size × Inoculum size | 1 | 3.7382 | 3.73816 | 0.000 | 0.000 |
| Moisture level × Moisture level | 1 | 2.9145 | 2.91453 | 0.000 | 0.000 |
| Incubation Period × IncubationPeriod | 1 | 0.0527 | 0.05272 | 0.000 | 0.000 |
| **2-Way Interaction** | 9 | 24.4761 | 2.71956 | 0.000 | 0.000 |
| pH × Temperature | 1 | 1.6593 | 1.65935 | 0.000 | 0.000 |
| pH × Inoculum size | 7 | 0.0138 | 0.01380 | 0.000 | 0.000 |
| pH × Moisture Level | 1 | 4.1829 | 4.18292 | 0.000 | 0.000 |
| pH × Incubation Period | 1 | 2.9296 | 2.92965 | 0.000 | 0.000 |
| Temperature × Inoculum Size | 1 | 0.5634 | 0.56339 | 0.000 | 0.000 |
| Temperature × Moisture Level | 1 | 0.5634 | 1.43103 | 0.000 | 0.000 |
| Temperature × Incubation Period | 1 | 0.4733 | 0.47331 | 0.000 | 0.000 |
| Inoculum Size × Moisture Level | 1 | 0.8363 | 0.83635 | 0.000 | 0.000 |
| Inoculum Size × Incubation Period | 1 | 3.5240 | 3.52397 | 0.000 | 0.000 |
| **Error** | 0 | | | | |
| **Total** | 19 | 87.8011 | | | |

Model Summary. S: 0.0; R-sq: 100.00%; R-sq (adj): 0.00; R-sq (pred): 0.00.

### 2.6. Lipase Purification

A 250mL volume of the crude enzyme with the highest lipase activity of 3623U/250mL was extracted from the best-optimized RSM trials and was subjected to further treatment for purification. With the 70% ammonium sulfate precipitation, partial purification of crude lipase was performed, which has a specific activity of 34.291 U/mg.

After ammonium sulfate precipitation, the saturated enzyme was dialyzed for more purification. With the change in buffer after regular intervals of time, dialysis removed the extra salts from the enzyme. When the enzyme assay was performed after dialysis, it was observed that specific enzyme activity increased to 48.03 U/mg. It was purified to homogeneity by column chromatography, and activity was increased to 132.72 U/mg. A purification fold of 5.07 with a 1.2 yield was obtained after all purification steps. Previously, for the purification of lipase from *Aspergillus niger* NC1M 1207, ammonium sulfate was used to precipitate the protein in the broth (90% saturation). The precipitated enzyme was dissolved in a small amount of glycine-NaOH buffer (50 mM, pH 9.0), then dialysis was undertaken. With an overall yield of 54% and a specific activity of 1373 IU/mg, the enzyme was purified approximately 150 folds. Sephacryl S-100 gel filtration chromatography was used for final purification [41]. Effect of salt concentration used in the precipitation of enzyme recovery is shown in Figure 23.

**Figure 23.** Effect of salt concentration used in the precipitation of enzyme recovery.

### 2.7. Protein Estimation

Different volumes of lipase were extracted at the different purification stages, and total enzyme activity for the whole volume was calculated from enzyme activity using the Bradford assay. By following the BSA, the protein content that was obtained was further used to elute the total enzyme protein content, even when the volume was in large excess. These values were very helpful as they provided information about lipase-specific activity and the increase in purification fold after every step. Enzyme-specific activity increased more and more with every purification step. Our results are consistent with those of Sethi et al. [42], who reported that lipase from *Aspergillus terreus* was purified using 80% $(NH4)_2SO_4$ precipitation followed by Sephadex G-100 with a purification fold of 2.56%. Okunwaye et al. [43] reported that lipase from Raphia mesocarp was purified using 80% $(NH4)_2SO_4$ precipitation followed by gel filtration chromatography, with a purification fold of 5.79% [44]. Purification summary of lipase production is shown in Table 3.

**Table 3.** Purification summary of lipase production

| Purification Step | Volume (mL) | Total Enzyme Activity (U/mL) | Total Protein Content (U/mg) | Specific Activity (mg) | Purification Fold |
|---|---|---|---|---|---|
| Crude Lipase | 250 | 3623 | 138.55 | 26.15 | 1 |
| (NH$_4$)$_2$SO$_4$ | 100 | 1635 | 47.68 | 34.29 | 1.3 |
| Dialyzed | 25 | 411 | 8.562 | 48.03 | 1.83 |
| Pure lipase | 10 | 224 | 1.69 | 132.72 | 5.07 |

*2.8. Characterization of Fungal Lipase*

Characterization of fungal lipase was performed by studying the effect of temperature and pH on lipase activity.

2.8.1. Effect of Temperature on Lipase Activity

Enzyme purification suggested lipase activity at different temperatures, and maximum activity was recorded at 40°C. Activity increased up to this temperature, but a further increase in temperature resulted in a decrease in enzyme activity. Various studies also demonstrated maximum lipolytic activity within the range of 37–45°C [45]. Temperature characterization graph for lipase activity is shown in Figure 24.

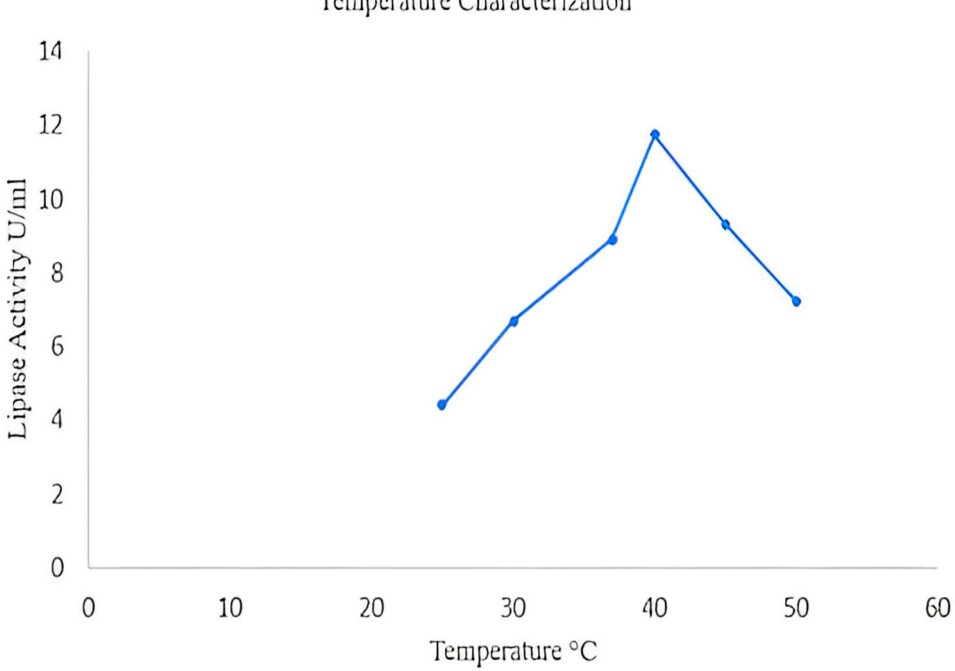

**Figure 24.** Temperature characterization graph for lipase activity.

2.8.2. Effect of pH on Lipase Activity

Enzyme purification suggested lipase activity at different pH, but maximum activity was recorded at pH 8. Activity was also observed at lower and higher pH, but it was less than the activity at pH 8. Various previous studies have also demonstrated that Aspergillus shows maximum enzyme activity in the pH range of 6–10. The pH characterization graph for lipase activity is shown in Figure 25.

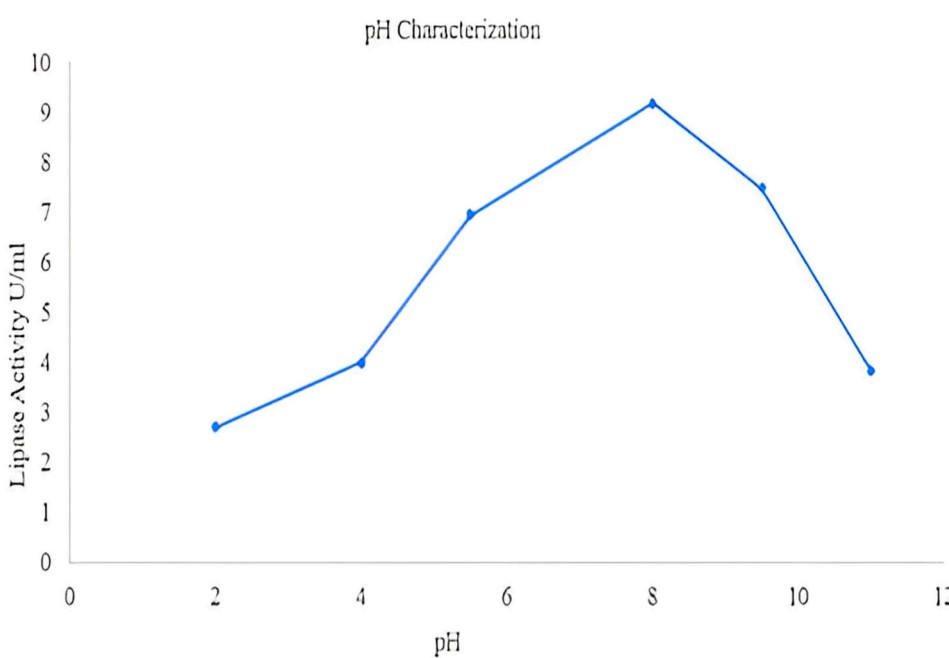

**Figure 25.** pH characterization graph for lipase activity.

## 3. Materials and Methods

### 3.1. Substrate Collection

In this study, all substrates used were collected from the surroundings of Gujrat, Pakistan. All substrates were dried by sunlight and then in an oven at 60°C. Then, these substrates were ground into small pieces and placed in a plastic jar to prevent moisture [46].

### 3.2. Strain for Lipase Production

Pure culture of *Aspergillus niger* was collected from the Department of Biochemistry and Biotechnology of the University of Gujrat. Potato dextrose agar (PDA) was used for the growth of all fungi. Therefore, for the growth of *Aspergillus niger*, PDA liquid and PDA solid slant were prepared. First, PDA media were prepared by adding 5 g glucose, 0.05 g dipotassium hydrogen phosphate ($K_2HPO_4.7H_2O$), 0.125 g hydrous magnesium sulfate ($MgSO_4.7H_2O$), 0.05 g ammonium sulfate (($NH_4)_2SO_4$), 0.125 g calcium chloride ($CaCl_2$), and 5 g agar in 250 mL distilled water. It was autoclaved and transferred to Petri dishes. Then, the spores of *Aspergillus niger* were transferred to Petri plates, and tubes and slants were incubated for sporulation. After sporulation, the spores were placed in a refrigerator at 37°C for future use. Fungal cultures were grown again after 15 days so that their viability could be maintained.

### 3.3. Screening of Fungi for Lipase Production

To confirm the ability of the fungal strain for the production of lipase, it was grown with inducers (sesame oil, mustard oil, Tween 80, and olive oil) in freshly prepared potato dextrose agar [40].The hydrolysis of inducers was used to test the fungal strain for lipase production. It is a qualitative agar plate assay used to determine a strain's ability to produce lipase. A lipolysis zone (clear zone) was observed around fungal colonies after incubation. The fungus showed a clear zone around the growth to release lipase, which causes cleavage of inducers present in the medium's oil and Tween 80 [47].

### 3.4. Vegetative Culturing

For the vegetative growth of *Aspergillus niger*, inoculum media were prepared. In Erlenmeyer flasks, PD media were prepared for the inoculum having the following composition: glucose 5 g, calcium chloride 0.125 g, dipotassium hydrogen phosphate 0.05, magnesium sulfate 0.125 g, and ammonium sulfate 0.05 g. The flask was then tightly closed

with a cotton plug. The sample was autoclaved at 15 lb/in2 and 121 °C for sterilization prior to fungal transfer. These flasks were then incubated for 72 h at 37 °C and 120 rpm in a water bath [48].

### 3.5. Screening of Substrate

A 5 g amount of every substrate was taken in flasks, moisturized with 5 mL distilled water, and autoclaved. Then, in the aseptic environment, 3 mL of the inoculum containing freshly prepared fungal spores was poured into each flask. These flasks were then incubated at 37 °C in a static incubator for 3 days [12].

### 3.6. Enzyme Extraction

For the extraction of the crude enzyme, 50 mL of distilled water was poured into each flask, then they were centrifuged for 30 min at 160 rpm. Whatman No.1 filter paper was used for the filtration of experimental mass, and then the sample was centrifuged for 20 min at a temperature of 40°C and 4400 rpm. Pellets and supernatant were obtained, pellets were preserved, and the supernatant was further used to check the lipolytic action of the crude enzyme [49].

### 3.7. Enzyme Assay

The spectrophotometric assay was used for the estimation of lipase activity by measuring the hydrolysis of p-nitrophenyl palmitate of our enzyme at a wavelength of 410 nm [43]. A 2.5 mL volume of 0.1 M Tris-Cl buffer, pH 8.2, and 2.5 mL of p-nitrophenyl palmitate substrate solution were taken into a 15 to 20mL test tube for each lipase activity assay. We also prepared one additional test tube for a blank reagent and added 1 mL of water instead of enzyme solution into the blank reagent. For the enzyme assay, 2.5 mL of 0.1M Tris-Cl and 2.5 mL of 420μM of para-nitrophenyl palmitate were taken in a test tube. A 1 mL volume of crude enzyme extracted after centrifugation was added to the test to start the reaction. The reaction mixture was transferred directly into a quartz cuvette to check its absorbance at 410 nm by using a spectrophotometer. Absorbance was recorded every 30 s for 15 min. The p-nitrophenol standard curve was used to measure lipase activity from absorbance. One unit of lipase activity was defined as the amount of lipase that released 1 μ mol of free p-nitro phenol in one minute [50].

### 3.8. Standard Curve of Para-Nitrophenol

To make the standard curve, a stock solution of 0.5 mM para-nitrophenol with 0.05 to 0.50 mL concentration was taken in ten 15 to 20 mL test tubes, and each was diluted with 0.1 M Tris-Cl buffer having pH 8.2 and made into a final solution of 5 mL. Para-nitrophenol acted as substrate and helped in the allocation of standard values for measuring the activity of lipase obtained by response surface methodology, and substrate activity was drawn using this standard curve as a reference for the following results.

### 3.9. Optimization of Lipase by Response Surface Methodology

After the selection of substrate and strain, according to the potential different conditions of enzymes, different physical and nutritional parameters, such as pH (2–11), time period (1–8 days), temperature (30–52 °C), inoculum size (1–8 mL), substrate concentration (1–8 g), and moisture (20–125%), were optimized using RSM to obtain maximum activity.

RSM trials for the optimization of different parameters that influence fungal lipase production are shown in Table 4.

**Table 4.** Optimization via RSM trials.

| Sr # | Substrate Size (g) | pH | Temperature (°C) | Inoculum Size (mL) | Moisture Level (%) | Incubation Time (Days) |
|---|---|---|---|---|---|---|
| 1 | 4 | 2 | 45 | 5 | 60% | 5 |
| 2 | 4 | 2 | 37 | 5 | 60% | 3 |
| 3 | 2 | 2 | 40 | 2 | 100% | 4 |
| 4 | 2 | 2 | 45 | 3 | 20% | 6 |
| 5 | 6 | 5.5 | 45 | 3 | 100% | 3 |
| 6 | 8 | 5.5 | 40 | 2 | 40% | 3 |
| 7 | 4 | 5.5 | 40 | 4 | 60% | 4 |
| 8 | 6 | 5.5 | 37 | 2 | 100% | 8 |
| 9 | 8 | 8 | 40 | 2 | 20% | 8 |
| 10 | 6 | 8 | 40 | 3 | 40% | 3 |
| 11 | 4 | 8 | 45 | 2 | 60% | 6 |
| 12 | 2 | 8 | 45 | 1 | 100% | 8 |
| 13 | 4 | 9.5 | 30 | 8 | 60% | 1 |
| 14 | 4 | 9.5 | 30 | 3 | 125% | 4 |
| 15 | 4 | 9.5 | 40 | 1 | 60% | 6 |
| 16 | 7 | 9.5 | 50 | 4 | 60% | 8 |
| 17 | 4 | 11 | 30 | 5 | 50% | 1 |
| 18 | 4 | 11 | 30 | 5 | 60% | 3 |
| 19 | 1 | 11 | 52 | 3 | 60% | 4 |
| 20 | 4 | 11 | 45 | 3 | 60% | 8 |

*3.10. Preparation of Standard Curve of Bovine Serum Albumin (BSA)*

To obtain the original results, different BSA concentrations were made to use as a reference for the allocation of enzyme activity at various stages of purification, along with percentage yield and purification fold. In a test tube containing the enzyme, 5 mL of Bradford reagent was added, and absorbance was monitored at 595nm between absorbance (*y*-axis) and concentration (x-axis) [51].

*3.11. Protein Content Determination*

For the determination of protein content from crude, completely and partially purified enzyme extractions and BSA were used [52].The concentration of total protein in a sample is determined using the Bradford protein assay. According to the assay's basic principle, protein molecules that bond to Coomassie dye in an acidic medium change their color from brown to blue. The protein–dye complex is formed when the basic amino acid residues arginine, lysine, and histidine are present in sufficient amounts [53].

*3.12. Protein Estimation*

From the known concentrations of BSA, the standard curve was made, which was further used for the quantification of protein.

$$\text{Protein in mg/mL} = \text{standard factor} \times \text{absorbance}$$

$$\text{Standard factor} = \text{slope} \times \text{dilution} \times \text{volume of sample}$$

*3.13. Bradford Reagent*

Bradford reagent is prepared by the dissolution of Coomassie brilliant blue G-250 in 100 mL of 85% phosphoric acid and 50 mL of 95% ethanol and made a final volume of up to 1 L. Afterwards, it is filtered through muslin cloth and stored at 40 °C for reuse [46]. If samples are not readily soluble in the color reagent, then 1M sodium hydroxide is used. The color of the Bradford reagent should be white. Repeated filtrations are used to eliminate blue components [54].

### 3.14. Bradford Assay

A 50μLvolume of crude, including partially and completely purified enzyme extractions along with 1mLof freshly prepared BSA, was taken in Eppendorf tubes. After mixing our sample, absorbance was taken at 595 nm [39].The dye's more anionic blue form, which binds to protein, has a maximum absorbance at 590 nm. The amount of dye in the blue ionic form can thus be used to estimate the amount of protein. This is usually accomplished by measuring the solution's absorbance at 595 nm [55].

### 3.15. Large Scale Enzyme Production

Crude enzyme extractions were prepared in large amounts after the selection of optimized conditions. The experimental mass was harvested and then filtered using Whatman filter paper, and biomass cell debris free supernatant, which was our crude enzyme, was extracted and then stored at 40 °C for the following trials.

### 3.16. Purification of Lipase

Crude enzyme purification has the following steps: ammonium sulfate precipitation, dialysis, and column chromatography [56].

### 3.16.1. Ammonium Sulfate Precipitation

Ammonium sulfate was precipitated with crude enzyme in two steps: salting in and salting out [47]. A 1mLvolume of crude enzyme along with different concentrations (10 to 100%) of ammonium sulfate was taken in 10 falcon tubes to attain the saturation point. All falcon tubes were frozen at 40 °C overnight and then centrifuged for 20 min at 40 °C and 4400rpm. The supernatant was extracted for the enzyme assay. By dissolving the ammonium sulfate up to a saturation of 70% with stirring and then storing it at 40 °C overnight, 250 mL of crude enzyme extract was purified. Both supernatant and pellets were preserved. The supernatant was used for enzyme assay.

### 3.16.2. Dialysis

For enzyme purification, pellets were used as they were placed in a dialysis bag that was placed in phosphate buffer inside a 500mL beaker. Then, they were placed in a shaking incubator overnight. Buffer was changed after regular intervals [57]. Buffer with different pH extracted the ammonium sulfate from our enzyme solution and was placed inside the dialysis bag, resulting in its increased purification.

### 3.16.3. Column Chromatography

For enzyme purification to the homogeneity level, column chromatography was used. It is one of the most commonly used techniques for the column that was first packed with sand and Sephadex G-100. It was washed with water and then with phosphate buffer, and then 10 mL of the dialyzed enzyme was loaded in the column with a flow rate of 1mL/min. Twenty elusions were extracted, and for the evaluation of one lipase having maximum activity, an enzyme assay was performed [57].

### 3.17. Characterization of Fungal Lipase

Purified fungal lipase characterization was performed through kinetic studies by the evaluation of the effect of pH and temperature on lipase activity.

### 3.17.1. Effect of Temperature on Lipase Activity

To evaluate the temperature response, the purified enzyme fraction was assayed in the temperature range of 30–52 °C for maximum activity of lipase.

### 3.17.2. Effect of pH

For the optimization of the pH range (2–11), different buffers were prepared, i.e., 0.1 M citrate buffer of pH 6.8, M phosphate buffer (pH 6.0–8.0), sodium acetate buffer

(pH 2.0–5.0), and sodium carbonate buffer (pH 9.0–10.0). The activity of lipase in different pH buffers was determined by enzyme assay.

## 4. Conclusions

Fungal enzymes have gained significant importance in almost every field of life due to their numerous applications. They are used in many industries, such as dairy, agriculture, textiles, and pharmaceuticals. In this research, a novel strain of *Aspergillus niger* was used for the production of lipase on a naturally occurring substrate. Nine substrates were taken at the start of the experiment, and then their relative fungal enzyme production through solid-state fermentation was observed by absorbance against a mass spectrophotometer. Out of these nine substrates, guava leaves showed the maximum enzyme activity, which was then further used for optimization through RSM, purification, and characterization.

Various nutritional and physical parameters were optimized through RSM for the production of lipase. Maximum lipase activity of 12.52 U/mL was observed at pH 8, 40°C temperature, 6g of substrate, an inoculum size of 3mL, and an incubation period of 4 days. The enzyme was partially purified by 70% ammonium sulfate precipitation. Then, the partially purified enzyme was dialyzed in a dialysis bag overnight with continuous changes of buffer at regular intervals, which eluted the extra particles in the enzyme and increased its purification level. It was then purified further to homogeneity, and specific enzyme activity of 132.72 U/mg with a 5.07 purification fold was obtained by column chromatography using a sephadex G-100 gel filtration column.

This study showed that *Aspergillus niger* can be grown on cheap media such as guava leaves, and the production of lipase made it economical to be further used in industrial applications. Using dry waste guava leaves for industrial purposes can also reduce pollution significantly, so it is for the benefit of human beings. This enzyme is even stable at higher temperatures, and thermostability is a necessary requirement of any enzyme to be used in industries. The outcomes of this study showed the excellent potential of *Aspergillus niger* for the proper treatment of many wastes as these are utilized as substrates for the growth of fungi and are excellent producers of alkaline lipase.

**Author Contributions:** Conceptualization, M.Z. and Z.A.; methodology, U.A.; software, M.Z.; validation, W.K., M.A.R. and Z.A.; formal analysis, U.A.; investigation, N.u.A.; resources, S.H.; data curation, M.Z.; writing—original draft preparation, U.A.; writing—review and editing, W.K.; visualization, F.A.; supervision, Z.A.; project administration, U.A. and A.A.-F.; funding acquisition, A.A-F., H.N.M. and H.A.R.A.E. All authors have read and agreed to the published version of the manuscript.

**Funding:** The Deanship of Scientific Research at King Khalid University funded this work through the Small Group Research Project under grant number (RGP.1/355/44).

**Data Availability Statement:** Not applicable.

**Acknowledgments:** The authors extend their appreciation to the Deanship of Scientific Research at King Khalid University for funding this work through the Small Group Research Project under grant number (RGP.1/355/44).

**Conflicts of Interest:** The authors declare no conflict of interest.

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
