# Peer review of "Bioprocessing and Screening of Indigenous Wastes for Hyper Production of Fungal Lipase"

_catalysts, doi:10.3390/catal13050853_

Round 1

Reviewer 1 Report

This work screened 8 lignolitic substrates for the lipase production, among which, Guava leaves showed maximum activity 9.1 U/ml from Aspergillus niger by using the solid-state fermentation. The optimized culture conditions were obtained by the authors, namely 4 days of incubation, at pH 8, temperature 40℃, 3 ml inoculum size, 20% moisture content and 6g substrate concentration. This work obtained a purification fold of 5.07 after all purification steps. The manuscript is in general well written and the method development process is described in detail. However, the following issues have to be addressed before this manuscript is suitable for publication.

1. The section Introduction may be a little lengthy, and this part of content should be compressed.

2. More experimental details should be provided in the sections 2.3 and 2.11-2.14.

3. More data analysis should be added regarding sections 3.2 and 3.5-3.7.

4. There are few contents in the section Discussion, please expand this section.

5. The section Conclusions may be a little lengthy, and this part of content should be compressed.

Author Response

This work screened 8 lignolitic substrates for the lipase production, among which, Guava leaves showed maximum activity 9.1 U/ml from Aspergillus niger by using the solid-state fermentation. The optimized culture conditions were obtained by the authors, namely 4 days of incubation, at pH 8, temperature 40℃, 3 ml inoculum size, 20% moisture content and 6g substrate concentration. This work obtained a purification fold of 5.07 after all purification steps. The manuscript is in general well written and the method development process is described in detail. However, the following issues have to be addressed before this manuscript is suitable for publication.

  1. The section Introduction may be a little lengthy, and this part of content should be compressed.

Answer: The section introduction has been modified and the corection has been done.

  1. More experimental details should be provided in the sections 2.3 and 2.11-2.14.

Answer : .

2.3. Screening of fungi for lipase production

To confirm the ability of fungal strain for the production of lipase, it was grown with inducers (sesame oil, mustard oil, tween 80 and olive oil) in freshly prepared Potato Dextrose Agar [40]. The hydrolysis of inducers was used to test fungal for lipase production. It is a qualitative agar plate assay used to determine a strain's ability to produce lipase. A lipolysis zone (clear zone) was observed around fungal colonies after incubation. Fungus showed a clear zone around the growth to release lipase, which causes cleavage of inducers present in the medium's oil and Tween 80.

2.11. Protein content determination

For the determination of protein content from crude, completely purified, partially purified and BSA was used [45]. The concentration of total protein in a sample is determined using the Bradford protein assay. According to the assay's basic principle, protein molecules that bond to Coomassie dye in an acidic medium change their color from brown to blue. The protein-dye complex is formed when the basic amino acid residues arginine, lysine, and histidine are present in sufficient amounts.

2.12. Protein estimation

From the known concentrations of BSA standard curve was made which was further used for the quantification of protein.

Protein in mg/ml = standard factor x absorbance

Standard factor = slope x dilution x volume of sample

2.13. Bradford reagent

Bradford reagent is prepared by dissolution of coomassive brilliant blue G-250 in 100 mL of 85% phosphoric acid and 50 mL of 95% ethanol and made a final volume of up to one liter. Afterwards, it is filtered through muslin cloth and stored at 40 0C for reuse [46]. If samples are not readily soulube the colour reagent then 1M sodium hydroxide is used.The color of Bradford should be white. Repeated filtrations are used to eliminate blue components.

2.14. Bradford assay

50µl of crude including partially and completely purified enzyme extractions along with 1 ml freshly prepared BSA were taken in the Eppendorf tubes. After mixing our sample absorbance was taken at 595nm [39]. The dye's more anionic blue form, which binds to protein, has a maximum absorbance at 590 nm. The amount of dye in the blue ionic form can thus be used to estimate the amount of protein. This is usually accomplished by measuring the solution's absorbance at 595 nm.

  1. More data analysis should be added regarding sections 3.2 and 3.5-3.7.

Added with references.

3.2. Substrate screening

Eight substrates (rice bran, wheat bran, guava leaves, peanut husk, corn husk, reed grass, sugarcane beggase) were inoculated with the spores from pure inoculum media to evaluate the best substrate for the production of lipase. The Substrate that proved to be best was guava leaves as maximum lipolytic activity was obtained from all other substrates. Previously an experimental design was used to incorporate sugarcane bagasse, wheat bran and, Soyabean bran into the lipase production process by Aspergillus sp. and Penicillium sp

  1. There are few contents in the section Discussion, please expand this section.

    For more better understanding Results and Discussion have modified.

  1. The section Conclusions may be a little lengthy, and this part of content should be compressed.

     Done

Reviewer 2 Report

The introduction section is too long and often repetitive. There also some awkward statements. The authors should rewrite it in a more comprehensive and crisp manner in order to summarize the key findings on lipases and, within the scope of their work, what is missing and how their work may fill in gaps. The authors should avoid repetition and excessive adjective and should be careful with scientific soundness of their statements. For recent reviews on lipases that may be useful, check, e.g., https://doi.org/10.1186/s12934-020-01428-8, https://doi.org/10.3389/fbioe.2017.00016, https://doi.org/10.1139/cjm-2016-044. Maybe the authors should reorganize the introduction as to identify concisely key aspects of lipase and how the apporach based on SSF may be advantageous

“Lipases are simply triacylglycerol acyl hydrolases (EC 3.1.1.3)” should be “Lipases are triacylglycerol acyl hydrolases (EC 3.1.1.3)”; “These All flasks”, either these flasks or all flasks, but not both; “50ml of distilled water was poured in each”, in each what? Please specify. “mints” should be minutes. There are many other similar glitches throughout the text, the authors should carefully read prior to uploading the manuscript.

Mind the correct use of tenses in the different parts of the manuscript.

The information conveyed in Tables 1 and 2 should be transferred to text.

“diazanium sulphate” use ammonium sulphate instead

Figures 1, 2, 4, 7, 8 should be deleted; they convey no relevant information.

Sections 2.8 and 2.10: the information is pretty much well-established knowledge, please be more concise. Accordingly, figures 3 and 5 are pretty much standard, they could either be deleted or, at the most, provided as supplementary material.

The authors should provide more details on the experimental procedure to evaluate enzyme activity under different temperatures and pH environments (including the number of replicates to establish statistical significance). Also, at temperature close to 50 ºC evaporation may be an issue unless capped vessels are used . Additionally, phosphate buffer has buffering capability around 5 to 8, so how could it be used as “0.1M phosphate buffer of pH 2, 4, 5.5, 9.5, 8 and 11”? This is not correct

How many replicates were performed during substrate screening runs?

The legend of Figure 9 must be improved to that the reader may easily grasp its meaning without having to go through all the text

The results depicted in figuews 10 and 32 seem pretty much the same, athough the strucure of the graphic is different. Does figure 10 also refer to the effect of temperture on enzyme activity? As it would be pointless to state here that production was optimal at 40 ºC in one factor at a time run if afterwards RSM is to be performed!

The authors addressed plenty of independent variables, what software was used to establish the equation relating dependent and independent variables?

How come F-value and p-value are always 0.000 in Table 2.2.

Did the authors perform an extra set of runs to validate the model? It is not clear from the data presented.

The y axis in figure 31 should provide enzyme activity (which is determined at 410 nm)

Table 5 must be improved, more information is needed to understand what is described therein.

The discussion section is very poor, either the authors combine results and discussion (and remove the present discussion section), or they improve discussion section by debating here the results obtained and how they can be interpreted, also taking into consideration the literature

In the conclusions, the authors state that “this enzyme is even stable at higher temperature and thermo stability is a necessary requirement of any enzyme to be used in industries” but in their work they did not perform enzyme stability studies, so conclusions are not supported by data

Author Response

The introduction section is too long and often repetitive. There also some awkward statements. The authors should rewrite it in a more comprehensive and crisp manner in order to summarize the key findings on lipases and, within the scope of their work, what is missing and how their work may fill in gaps. The authors should avoid repetition and excessive adjective and should be careful with scientific soundness of their statements. For recent reviews on lipases that may be useful, check, e.g., https://doi.org/10.1186/s12934-020-01428-8, https://doi.org/10.3389/fbioe.2017.00016, https://doi.org/10.1139/cjm-2016-044. Maybe the authors should reorganize the introduction as to identify concisely key aspects of lipase and how the apporach based on SSF may be advantageous

“Lipases are simply triacylglycerol acyl hydrolases (EC 3.1.1.3)” should be “Lipases are triacylglycerol acyl hydrolases (EC 3.1.1.3)”; “These All flasks”, either these flasks or all flasks, but not both; “50ml of distilled water was poured in each”, in each what? Please specify. “mints” should be minutes. There are many other similar glitches throughout the text, the authors should carefully read prior to uploading the manuscript.

Answer: Thank you so much for your valuable suggestions. All the Corrections have been done. Please see the revised version.

Mind the correct use of tenses in the different parts of the manuscript.

Answer: We have corrected them according to your kind comments. Please see the revised version.

The information conveyed in Tables 1 and 2 should be transferred to text.

Answer: Thank you for letting us know. We have corrected them according to your kind comments. Please see the revised version.

“diazaniumsulphate” use ammonium sulphate instead

Answer : We have corrected them according to your kind comments. Please see the revised version.

Figures 1, 2, 4, 7, 8 should be deleted; they convey no relevant information.

Answer: We highly appreciate your advice. We have deleted them. Please see the revised version.

Sections 2.8 and 2.10: the information is pretty much well-established knowledge, please be more concise. Accordingly, figures 3 and 5 are pretty much standard, they could either be deleted or, at the most, provided as supplementary material.

Answer: Thank you for your positive comments. We highly appreciate your advice. We have deleted them. Please see the revised version.

The authors should provide more details on the experimental procedure to evaluate enzyme activity under different temperatures and pH environments (including the number of replicates to establish statistical significance). Also, at temperature close to 50 ºC evaporation may be an issue unless capped vessels are used . Additionally, phosphate buffer has buffering capability around 5 to 8, so how could it be used as “0.1M phosphate buffer of pH 2, 4, 5.5, 9.5, 8 and 11”? This is not correct

For the optimization of pH range (2-11) different buffers were prepared i.e 0.1M citrate buffer of pH 6.8,M phosphate buffer (pH 6.0-8.0),, Sodium acetate buffer (pH 2.0-5.0), and sodium carbonate buffer (pH 9.0-10.0) . The Activity if lipase in different pH buffers was determined by enzyme assay.

How many replicates were performed during substrate screening runs?

Answer: Thanks for your valuable suggestions. Triplicates were performed during substrate screening runs.

The legend of Figure 9 must be improved to that the reader may easily grasp its meaning without having to go through all the text

Answer: Improved. After deletion of some figures now its added as Figure 4.

Figure 4. Biomass screening of Substrates to evaluate the best subrate for Lipase production.

The results depicted in figuews 10 and 32 seem pretty much the same, athough the strucure of the graphic is different. Does figure 10 also refer to the effect of temperture on enzyme activity? As it would be pointless to state here that production was optimal at 40 ºC in one factor at a time run if afterwards RSM is to be performed!

Answer: Figure 10 removed and statement is corrected.

The authors addressed plenty of independent variables, what software was used to establish the equation relating dependent and independent variables?

Statistical Analysis was performed using the software minitab 17. RSM readings were analyzed to measure the fitness of model by R2 value.

How come F-value and p-value are always 0.000 in Table 2.2.

Answer: Thanks for your kindness. It was seen when regression analysis was performed. And it shows the R squre value 1 which indicates the model fitness

Did the authors perform an extra set of runs to validate the model? It is not clear from the data presented.

After selecting the peak value substrate then RSM was run to extract the optimum parameters for this experiment. Extra sets of parameters Temp and pH were run to validate the charaterization of lipase

Table 5 must be improved, more information is needed to understand what is described therein.

Answer: Thanks for your suggestions.  We have revised Table 5.

Table 5. Purification summary of lipase production

Purification Step

Volume (ml)

Total Enzyme Activity (U/ml)

Total Protein Content ( U/mg)

Specific Activity (mg)

Purification Fold

Crude Lipase

250

3623

138.55

26.15

1

(NH4)2SO4

100

1635

47.68

34.29

1.3

Dialyzed

25

411

8.562

48.03

1.83

Pure lipase       

10

224

1.69

132.72

5.07

The discussion section is very poor, either the authors combine results and discussion (and remove the present discussion section), or they improve discussion section by debating here the results obtained and how they can be interpreted, also taking into consideration the literature

Answer: Thanks for your suggestions. Results and discussion have been modified.

In the conclusions, the authors state that “this enzyme is even stable at higher temperature and thermo stability is a necessary requirement of any enzyme to be used in industries” but in their work they did not perform enzyme stability studies, so conclusions are not supported by data

Answer: We highly appreciate your valuable comments. Basically the aim of this research was production and optimization of lipase. To perform enzyme stabilty extra approaches required but due to lack of apparatus facility we have focused on production of lipase from biomass

Reviewer 3 Report

The novelty and the quality of the manuscript are good and it does not need extensive improvement before publication. It is carefully organized and written. It is easy to follow it and contains clear comments and conclusions. 

In my opinion, this manuscript is very detailed and meticulous, it covers all the literature in the field with critical point of view. The topic have been completely covered and is well connected through the text. There is a significant  novelty in presented topic.  For all these reasons, I can recommend the acception of the manuscript after minor revision. Few minor suggestion are as follow:

1. I think that part about solid-state fermentation could be extended, more examples  should be added. This would be valuable for later publication citation.

2. The superiority of  the optimisation by response surface methodology  than other optimisation methods should be more emphasized.

3. The manuscript should be extended in scientific discussion. The authors presented their results and compared to some works, but did not present explanations for the reasons to reach these results.

4. Not all of the described results are covered in the discussion section.

5. No all information was given about chemistry of lipases.

Author Response

The novelty and the quality of the manuscript are good and it does not need extensive improvement before publication. It is carefully organized and written. It is easy to follow it and contains clear comments and conclusions. 

In my opinion, this manuscript is very detailed and meticulous, it covers all the literature in the field with critical point of view. The topic have been completely covered and is well connected through the text. There is a significant  novelty in presented topic.  For all these reasons, I can recommend the acception of the manuscript after minor revision. Few minor suggestion are as follow:

  1. I think that part about solid-state fermentation could be extended, more examples  should be added. This would be valuable for later publication citation.

More information about solid state fermentation has added.

Solid state fermentation is now used in a variety of newly developed products such as bioactive compounds and organic acids, new trends such as bioethanol and biodiesel as alternative energy sources, and biosurfactant molecules with environmental purposes such as valorizing unexploited biomass. The nature of specific microorganisms and substrates influences the success of applying solid state fermentation to a specific process. A comprehensive list of microorganisms capable of growing in a solid state fermentation is presented, including fungi such as Aspergillus or Penicillum for antibiotic production, Rhizopus for bioactive compounds

More examples of SSF are added in introduction section.

  1. The superiority of  the optimisation by response surface methodology  than other optimisation methods should be more emphasized.

Superiority of optimization by RSM is added in section 3.3

  1. The manuscript should be extended in scientific discussion. The authors presented their results and compared to some works, but did not present explanations for the reasons to reach these results.

Reference based results have added with comparison of previous studies.

  1. Not all of the described results are covered in the discussion section.

Results and Discussion have modified now.

  1. No all information was given about chemistry of lipases.

Lipases are triacylglycerol acyl hydrolases (EC 3.1.1.3) which are further categorized into monoglycerides, diglycerides, and fatty acids such as serine hydrolases that break down triglycerides [7]. Lipase is a subclass of esterases and has a long chain of triacylglycerol, which has very less water solubility, and catalyzed reactions take place at the interface of lipid-water [9]. Lipases are very effective in catalyzing reactions taking place in aqueous and non-aqueous media owing to their high temperature, pH, and organic solvent stability. Lipase has a hydrophobic lid which is essential for its interfacial function [10]. Lipases are modified in such a way that they can work at the interfaces of biphasic structures, and this process is known as interfacial activation. The characteristic of the substrate can be done by micelle, aggregate, or monomolecular film formed by an ester molecule. Lipases catalyze esterification, transesterification, and inter-esterification reactions in the organic medium [11]. Lipase substrates are diversified and can be composed of phospholipids, neutral lipids, ether lipids, and lysophospholipids.

Round 2

Reviewer 2 Report

Some improvements were made but there are several relevant matters pending. Again, the introduction section is still too long and often repetitive. There also some awkward statements. The authors should rewrite it in a more comprehensive and crisp manner. Please,  summarize the key findings on lipases related to the scope of their work. Clearly identify what is missing and how their work may fill in gaps.

The authors should avoid repetition and excessive adjective and should be careful with scientific soundness of their statements, see for example “Lipase is a subclass of esterases and has a long chain of triacylglycerol, which has very less water solubility, and catalyzed reactions take place at the interface of lipid-water”

For recent reviews on lipases that may be useful, check, e.g., https://doi.org/10.1186/s12934-020-01428-8, https://doi.org/10.3389/fbioe.2017.00016, https://doi.org/10.1139/cjm-2016-044.

The authors are still advised should reorganize the introduction as to identify concisely key aspects of lipase and how the work performed may improve current knowledge

Figure 1(?). Standard curve of 4-nitrophenol should be deleted, it refers to an established method and does not add relevant information.

Section 2.11 to 2.14, the information conveyed is pretty much well established. In a previous review it was suggested that the authors should be more concise but the opposite was done. Pease summarize, quantification of protein through Bradford method is well knon and requires a paragraph or two at the most, not 4 sub-sections. Accordingly, Figure 2(?). Standard curve of bovine serum albumin must be deleted.

The authors use different buffers to access the effect of pH on lipase activity. Did they rule out any intrinsic effect of buffer composition, e.g., by evaluating enzyme activity at a given pH using different buffers?

Figure 3(?). Partial purification by ammonium sulfate precipitation, does not provide relevant information and must be deleted.

How come F-value and p-value are always 0.000 in Table 2.

Did the authors perform an extra set of runs to validate the model? It is not clear from the data presented.

Figure 2(?). Ammonium sulfate precipitation. The authors should improve the legend as to properly describe what is depicted in the graphic, i.e., effect of salt concentration used in the precipitation of enzyme recovery.

The authors added text stating “The same set of experiments was modeled using two different methods: response surface methodology (RSM) and artificial neural network (ANN). Statistical analyses revealed that both methods, RSM and ANN, can be used to accurately predict response, but the RSM (R2 = 0.9987) method was found to be slightly superior to the ANN model (R2 = 0.9973)” Does this refer to experiments performed in reference [51] and the authors aim to highlight that RSM can outmatch ANN? For a similar approach to that undertaken in the work submitted by the authors? This must be clarified!

In the conclusions, again the authors state that “This enzyme is even stable at higher temperature and thermo stability is a necessary requirement of any enzyme to be used in industries.” Yet the experimental runs do not include stability experiments, so this conclusion is not supported by the data.

The designation of microorganism must be italicized, and this is missing in some cases, please check

In p. 26 the chemical formula of ammonium sulfate is incorrect, the authors must address this

Author Response

Comments and Suggestions for Authors

Some improvements were made but there are several relevant matters pending. Again, the introduction section is still too long and often repetitive. There also some awkward statements. The authors should rewrite it in a more comprehensive and crisp manner. Please,  summarize the key findings on lipases related to the scope of their work. Clearly identify what is missing and how their work may fill in gaps.

The authors should avoid repetition and excessive adjective and should be careful with scientific soundness of their statements, see for example “Lipase is a subclass of esterases and has a long chain of triacylglycerol, which has very less water solubility, and catalyzed reactions take place at the interface of lipid-water”

Answer: Corrected.

For recent reviews on lipases that may be useful, check, e.g., https://doi.org/10.1186/s12934-020-01428-8, https://doi.org/10.3389/fbioe.2017.00016, https://doi.org/10.1139/cjm-2016-044.

Answer: Corrected.

The authors are still advised should reorganize the introduction as to identify concisely key aspects of lipase and how the work performed may improve current knowledge.

Answer :Introduction is revised and we tried to be more concise this time.

Figure 1(?). Standard curve of 4-nitrophenol should be deleted, it refers to an established method and does not add relevant information.

Answer: deleted.

Section 2.11 to 2.14, the information conveyed is pretty much well established. In a previous review it was suggested that the authors should be more concise but the opposite was done. Pease summarize, quantification of protein through Bradford method is well knon and requires a paragraph or two at the most, not 4 sub-sections. Accordingly, Figure 2(?). Standard curve of bovine serum albumin must be deleted.

Answer: Figure is deleted, but addition of data in section 2.11-2.14 is requirement of reviewer 1.

The authors use different buffers to access the effect of pH on lipase activity. Did they rule out any intrinsic effect of buffer composition, e.g., by evaluating enzyme activity at a given pH using different buffers?

Answer: The goal of this study was to determine the maximum enzyme activity, so different buffers with different pH values were used in the study to determine at which pH, enzyme activity is maximum. Many previous research papers have also supported the use of these buffers.

Figure 3(?). Partial purification by ammonium sulfate precipitation, does not provide relevant information and must be deleted.

Answer: deleted.

How come F-value and p-value are always 0.000 in Table 2.

Answer:

A reported p= 0.000 in computer output just means that results were highly significant (very unlikely to have occurred by chance alone). The true p-value is not 0.000. What happened is the actual p-value was less than 0.0005, and using built-in rounding rules, it was rounded down and reported as 0.000. whenever p-value is reported as 0.000 it means p-value is very small (less than 0.0005). in results it is reported as p < 0.001. The p-value of 0.000 means that results are highly significant. We know reporting p as 0.000 is generally frowned upon, because it  suggests there was absolutely no (zero) chance of getting these results if null hypothesis was true. But there is always some chance, however small. That’s why its reported as p < .001 instead of p=.000. but these reults are generated by  software results without any alterations. In text p value will be written as p < .001.

In general, an F statistic far into the lower tail suggestes that the fit is too good. F ratio is based on Analysis of Variance to compare group means: the means across all groups are exactly equal. For F to equal exactly 0, the explained variance would have to be exactly 0. In an ANOVA context, that would imply that the means in every group were exactly equal.

Did the authors perform an extra set of runs to validate the model? It is not clear from the data presented.

Answer : After selecting the peak value substrate then RSM was run to extract the optimum parameters for this experiment. Extra sets of parameters temperature and pH were run to validate the characterization of Lipase.

Figure 2(?). Ammonium sulfate precipitation. The authors should improve the legend as to properly describe what is depicted in the graphic, i.e., effect of salt concentration used in the precipitation of enzyme recovery.

Answer : Done as suggested

The authors added text stating “The same set of experiments was modeled using two different methods: response surface methodology (RSM) and artificial neural network (ANN). Statistical analyses revealed that both methods, RSM and ANN, can be used to accurately predict response, but the RSM (R2 = 0.9987) method was found to be slightly superior to the ANN model (R2 = 0.9973)” Does this refer to experiments performed in reference [51] and the authors aim to highlight that RSM can outmatch ANN? For a similar approach to that undertaken in the work submitted by the authors? This must be clarified!

Answer: This is requirement of reviewer 3 to add some data how RSM is superior to some other approach.

In the conclusions, again the authors state that “This enzyme is even stable at higher temperature and thermo stability is a necessary requirement of any enzyme to be used in industries.” Yet the experimental runs do not include stability experiments, so this conclusion is not supported by the data.

Answer: Basically the aim of this research was production and optimization of lipase. To check enzyme stability we used the progressive curve, its conversion of substrate to product in an assay. There should be a linear increase in conversion, followed by a plateau. We took the linear portion of the progress curve which is called the initial velocity, and measured the slope. Initial velocities stayed constant with time, and it showed our enzyme is stable under the reaction conditions even in high temperature.

The designation of microorganism must be italicized, and this is missing in some cases, please check.

Answer: corrected.

In p. 26 the chemical formula of ammonium sulfate is incorrect, the authors must address this.

Answer: corrected.
